# Generalizing while preserving monotonicity in comparison-based preference learning models

**Julien Fageot**[*]
Tournesol

**Peva Blanchard**[*]
Kleis Technology

**Gilles Bareilles**
CTU in Prague

**Lê-Nguyên Hoang**
Calicarpa, Tournesol

## Abstract

If you tell a learning model that you prefer an alternative $a$ over another alternative $b$, then you probably expect the model to be *monotone*, that is, the valuation of $a$ increases, and that of $b$ decreases. Yet, perhaps surprisingly, many widely deployed comparison-based preference learning models, including large language models, fail to have this guarantee. Until now, the only comparison-based preference learning algorithms that were proved to be monotone are the Generalized Bradley-Terry models [10]. Yet, these models are unable to generalize to uncompared data. In this paper, we advance the understanding of the set of models with generalization ability that are *monotone*. Namely, we propose a new class of Linear Generalized Bradley-Terry models with Diffusion Priors, and identify sufficient conditions on alternatives' embeddings that guarantee monotonicity. Our experiments show that this monotonicity is far from being a general guarantee, and that our new class of generalizing models improves accuracy, especially when the dataset is limited.

## 1 Introduction

Preference learning, sometimes known as *alignment*, has become central to machine learning. In particular, in recent years, there has been a growing interest to leverage comparative judgments to fine-tune AI models, using frameworks like *Reinforcement Learning with Human Feedback* (RLHF) [6] or *Direct Preference Optimization* (DPO) [31] in the context of language models, or linear models in the context of applications ranging from trolley dilemmas to food donation [1, 20]. These models are now deployed at scale. In parallel, learning preferences from comparisons with mathematical guarantees also fits in social choice theory, contributing to developing more transparent social medial, as advocated recently by "prosocial media" [39], with a direct application for collaborative scoring of social media content [16, 17], consensus-driven polling [34], or recommender system based on explicit preference such as [11], among others.

Yet, bizarre aspects of these preference learning algorithms are regularly observed. A common but striking observation is that, when updating a model based on a comparison judging that an item $a$ is preferable to an item $b$, the probability of $a$ can decrease [28, 33, 2]. In fact, most deployed learning algorithms fail to guarantee *monotonicity*: the probability or score of an item may be reduced after it was said to be better than another item.

Perhaps surprisingly, the root cause of this lack of monotonicity is *not* the nonlinearity of the models. In fact, we can expose this "bug" with a very basic example. Consider a linear two-dimensional model with a parameter $\beta \in \mathbb{R}^2$ to be learned: the scores of items $a$ and $b$ are $\beta^\top x_a$ and $\beta^\top x_b$, where $x_a$ and $x_b$ are (two-dimensional) vector embeddings. We are given a comparison that favors alternative $a$ over $b$, whose embeddings are $x_a = (1, 0)$ and $x_b = (2, 0)$. Since $x_{a1} < x_{b1}$, this comparison will push $\beta_1$ towards lower values. But since the score of $a$ according to the linear model is $\beta^T x_a = \beta_1$, this means that the score of $a$ will also decrease. Thus, including a comparison that favors $a$ over $b$

---

[*]Equal contribution.

39th Conference on Neural Information Processing Systems (NeurIPS 2025).

has decreased the score of $a$. This becomes all the more troubling when there exists an alternative $c$ with embedding $x_c = (0, 1)$, whose score remains unchanged.

This example questions whether preference learning algorithms can be trusted. A user who witnesses a surprising evolution of alternatives' scores as illustrated above might, understandably, prefer not to use such an algorithm. Worse yet, if they are nevertheless forced to use the algorithm, they could want to remove the data they previously provided, because the eventual learned model was deteriorated by their truthful data reporting. More generally, this may discourage users to report their preferences, and rather provide tactical preferences in the hope to steer the model towards their goal. This is reminiscent of tactical or "useful" strategies in voting systems. There exist classes of models which have a mathematical guarantee of monotonicity, such as (Generalized) Bradley-Terry models [10]; see also [27]. To the best of our knowledge, existing models with a guarantee of monotonicty fail to generalize: they cannot predict the score of items that have not been compared. Hence the following research problem.

*Can a generalizing comparison-based preference learning algorithm guarantee monotonicity?*

**Contributions.** Our first contribution is to identify a large class of preference learning algorithms that leverage both *comparisons* between alternative pairs, and *descriptive information* (embeddings) on individual alternatives, which we call *Linear Generalized Bradley-Terry with Diffusion Prior* (Definition 3). This class extends the (Generalized) Bradley-Terry models [10] by including a linear mapping of the embeddings—and priors on alternative similarities, thereby allowing preference *generalization* to yet uncompared alternatives.

As a second contribution, we provide conditions on the embeddings that guarantee that the learning algorithm behaves monotonically when new comparisons are provided. As discussed above, this property is highly desirable and yet hard to guarantee in practice. In particular, we propose a class of *diffusion embeddings* that guarantee monotonicity, and for which membership is easy-to-check. Interestingly, diffusion embeddings yield a very appealing interpretation as heat diffusion dynamics where comparisons play the role of heat pumps. A direct consequence is that categorical information (one-hot encoding embedding) yields a monotone learning algorithm. In particular, this class enables us to provide a positive answer to our research question.

Finally, we evaluate the statistical performance of our learning algorithms through numerical experiments. Our evaluations show that a linear model with good embeddings and diffusion priors outperforms the classical GBT model [10], in particular with limited amount of data.

**Related works.** Learning preferences from comparisons has a long history, dating back to Thurstone [37], Zermelo [40], and Bradley and Terry [3]. To handle inconsistent judgments, such algorithms define a probabilistic model of how latent scoring of alternatives are transformed into noisy comparisons. Their approach was adapted by [21] and [30] to model the selection of one preferred alternative out of several proposed ones; see also [23] and [22, Chap. 3].

While the Bradley-Terry model considers binary-valued comparisons, various authors have proposed extensions to real-valued comparisons, e.g., ranging the interval $[-1, 1]$. Historically, this started with the modeling of draws in games like chess [8]. More recently, [19] proposed the platform `Climpact` where users are given pairs of activities, and are asked to evaluate the comparative pollutions of two activities. They then develop a model based on a quadratic error to turn the comparisons into evaluations of the amounts of pollution of the individual activities. Their model was then generalized by [10] in a framework they call *Generalized Bradley-Terry* (GBT) to turn real-valued comparisons into scores. All these models however consider that each alternative's score is an independent latent variable to be learned. Thus, they fail to *generalize* to non-evaluated alternatives.

Independent of user-provided comparisons, alternatives usually come with descriptive information. A natural idea to generalize is then to model an alternative's score as a parametrized function of a vector embedding of the description of the alternative; see e.g., [24, 7, 41, 9, 12]. Recently, this trick has been widely used in the context of language models [38, 4], especially through algorithms like *Reinforcement Learning with Human Feedback* (RLHF) [6, 35], *Direct Preference Optimization* (DPO) [32], or *Generalized Preference Optimization* (GPO) [36], to name a few. In this paper, we restrict ourselves to *linear models*, where an alternative's score is assumed to be a linear function of their embedding. Application-wise, we focus on social choice applications, and rule out Supervised

FineTuning applications. Such linear models of preferences trained from comparative judgments have previously been studied and used, e.g. by [13, 26, 20].

The study of the mathematical guarantees of preference learning algorithms has only emerged recently. In particular, nonlinear models have been empirically shown to violate monotonicity properties [5, 28, 33]. While [2] proved that nonlinear models nevertheless provide a weak monotonicity guarantee they call *local pairwise monotonicity*, they also suggest that these models are unlikely to verify stronger forms of monotonicity. Conversely, and quite remarkably, [10] proved that the GBT model guarantees monotonicity for all GBT root laws. Our model extends GBT in several ways.

**Paper structure.** The rest of the paper is organized as follows. Section 2 introduces the formalism, formally defines monotonicity, and recalls the GBT model. Section 3 defines the linear GBT model with diffusion prior, and states our main results. Section 4 provides the main lines of the proofs of the main results. Section 5 reports on our experiments. Section 6 concludes.

## 2 Monotonicity of Scoring Models

In this Section, we set notations, formalize the notion of monotonicity, and recall the Generalized Bradley-Terry model.

### 2.1 Notations and operations on datasets

Consider a set $\mathcal{A}$ of $A$ alternatives. For simplicity, we let $\mathcal{A} \triangleq \{1, 2, \ldots, A\}$. The set $\mathfrak{R} \subseteq \mathbb{R}$ denotes the set of admissible comparison values, which we assume to be symmetric around zero, i.e. $r \in \mathfrak{R} \iff -r \in \mathfrak{R}$. In the classical Bradley-Terry model, we have $\mathfrak{R} = \{-1, +1\}$. The generalized Bradley-Terry model allows a wider variety of possible comparison values, for instance $\mathfrak{R} = [-1, 1]$, or $\mathfrak{R} = \mathbb{R}$ for the uniform and gaussian root laws. A comparison sample is defined as a triple $(a, b, r)$ where $a, b \in \mathcal{A}$ are two distinct alternatives, and $r \in \mathfrak{R}$. We assume $(a, b, r)$ and $(b, a, -r)$ to be equivalent, which we write $(a, b, r) \simeq (b, a, -r)$. By also having $(a, b, r) \simeq (a, b, r)$ (and the relation false otherwise), we obtain an equivalence relation. A dataset $\mathbf{D}$ is a list $\mathbf{D} : [N] \to \mathcal{A}^2 \times \mathfrak{R}$ of comparison samples. We write $\mathcal{D} \triangleq \bigcup_{N \in \mathbb{N}} (\mathcal{A}^2 \times \mathfrak{R})^N$ for the set of datasets, and $|\mathbf{D}|$ the length of a dataset $\mathbf{D} \in \mathcal{D}$. We now define four parameterized operations $\mathcal{D} \to \mathcal{D}$ on datasets.

**Exchange.** For any $n \in \mathbb{N}$, $\text{EXCHANGE}_n(\mathbf{D})$ is the dataset obtained from $\mathbf{D}$ by replacing, if it exists, the $n$-th entry $(a_n, b_n, r_n)$ with $(b_n, a_n, -r_n)$ All other entries are left unchanged. Assuming that preference learning algorithms should interpret these two comparison samples identically, this operation should not affect training.

**Shuffle.** For any $N \in \mathbb{N}$ and any permutation $\sigma$ of $[N]$, $\text{SHUFFLE}_{N,\sigma}(\mathbf{D})$ is the dataset obtained from $\mathbf{D}$ by reordering its $N$ first elements according to $\sigma$. Formally, if $|\mathbf{D}| \geq N$, then for all $n \in [N]$ we have $\text{SHUFFLE}_{N,\sigma}(\mathbf{D})_n = \mathbf{D}_{\sigma(n)}$. Otherwise, $\mathbf{D}$ is left unchanged. Assuming that preference learning algorithms should be invariant to shuffling, this operation should not affect training.

**Append.** For any comparison sample $(a, b, r)$, $\text{APPEND}_{a,b,r}(\mathbf{D})$ is the dataset obtained from $\mathbf{D}$ by appending $(a, b, r)$. Formally, we have $|\text{APPEND}_{a,b,r}(\mathbf{D})| = |\mathbf{D}| + 1$, and $\text{APPEND}_{a,b,r}(\mathbf{D})_{|\mathbf{D}|+1} = (a, b, r)$. All other entries are the same as in $\mathbf{D}$. An append is said to definitely favor $a'$ over $b'$ if it has parameters $(a, b, r) \simeq (a', b', \max \mathfrak{R})$. Note that if $\mathfrak{R}$ does not have a maximum, then no append definitely favors $a'$ over $b'$.

**Update.** For any $n \in \mathbb{N}$ and comparison $r \in \mathfrak{R}$, $\text{UPDATE}_{n,r}(\mathbf{D})$ is the dataset obtained from $\mathbf{D}$ by replacing the comparison of the $n$-th entry with $r$. The update is said to favor $a$ over $b$ if either (i) $(a_n, b_n) = (a, b)$ and $r \geq r_n$, or (ii) $(a_n, b_n) = (b, a)$ and $r \leq r_n$. In other words, it favors $a$ over $b$ if it acts on a comparison sample between $a$ and $b$, and modifies the comparison $r$ to further favor $a$.

## 2.2 Monotonicity

**Definition 1** (Favoring $a$). *An operation $o$ favors $a$ if $o$ is a composition of the operations (i)* EXCHANGE*, (ii)* SHUFFLE*, (iii)* APPEND *that definitely favor $a$ over some other alternative and (iv)* UPDATE *that favor $a$ over some other alternative. We write $\mathbf{D} \preceq_a \mathbf{D}'$ if there exists an operation $o$ that favors $a$ such that $\mathbf{D}' = o(\mathbf{D})$.*

The relation $\preceq_a$ is a preorder. Indeed, $\preceq_a$ is reflexive: any dataset $\mathbf{D}$ equals $o(\mathbf{D})$ with $o =$ UPDATE$_{n,r}$ with $n = 1$, and $r = r_1$. The relation $\preceq_a$ is transitive: if $\mathbf{D}_1 \preceq_a \mathbf{D}_2$ and $\mathbf{D}_2 \preceq_a \mathbf{D}_3$, then there exists operations $o_1$ and $o_2$ that favor $a$, such that $\mathbf{D}_1 = o_1(\mathbf{D}_2)$, and $\mathbf{D}_2 = o_2(\mathbf{D}_3)$; thus $\mathbf{D}_1 = o_1 \circ o_2(\mathbf{D}_3)$, where $o_1 \circ o_2$ is an operation that favors $a$ by Definition 1 so that $\mathbf{D}_1 \preceq_a \mathbf{D}_3$. Similarly, we define the preorder $\leq_a$ over $\mathbb{R}^A$ by $\theta \leq_a \theta'$ if $\theta_a \leq \theta'_a$ coordinate-wise. We can now formally define monotonicity.

**Definition 2** (Monotonicity). *The preference learning algorithm* ALG *is* monotone *when, for every alternative $a \in \mathcal{A}$,* ALG $: (\mathcal{D}, \preceq_a) \to (\mathbb{R}^A, \leq_a)$ *is monotone. Equivalently,* ALG *is monotone when, for every alternative $a \in \mathcal{A}$, $\mathbf{D} \succeq_a \mathbf{D}'$ implies* ALG$(\mathbf{D}) \geq_a$ ALG$(\mathbf{D}')$.*

**Remark 1.** *In the sequel, all preference learning algorithms that we will consider are* neutral *[25], i.e. they treat all alternatives symmetrically[2]. For such algorithms, the monotonicity for any single $a \in \mathcal{A}$ clearly implies that for all $a \in \mathcal{A}$.*

## 2.3 (Generalized) Bradley-Terry and monotonicity

Here we recall the probabilistic model of GBT [10], slightly adapting it to our dataset formalism.[3] Following Bradley and Terry [3], GBT defines a probabilistic model of comparisons given scores. Specifically, given two alternatives $a, b \in \mathcal{A}$ having scores $\theta_a$ and $\theta_b$, the probability of observing a value $r$ for the comparison of $a$ relative to $b$ is

$$p(r|\theta_{a\ominus b}) \propto f(r) \cdot \exp(r \cdot \theta_{a\ominus b}), \tag{1}$$

where $\theta_{a\ominus b} \triangleq \theta_a - \theta_b$ is the score difference between $a$ and $b$, and $f$ is the *root law*, a probability distribution on $\mathfrak{R}$ that describes comparisons when $a$ and $b$ have equal scores.

Given a dataset $\mathbf{D} = (a_n, b_n, r_n)_{n \in [N]}$ of $N$ independent observations following (1), and assuming a gaussian prior with zero mean and $\sigma^2$ variance for each alternative score $\theta_a$, $a \in \mathcal{A}$, the standard Maximum A Posteriori methodology results in the GBT estimator:

$$\mathrm{GBT}_{f,\sigma}(\mathbf{D}) \triangleq \arg\min_{\theta \in \mathbb{R}^{\mathcal{A}}} \frac{1}{2\sigma^2} \sum_a \theta_a^2 + \sum_{(a,b,r) \in \mathbf{D}} \Phi_f(\theta_{a\ominus b}) - r\theta_{a\ominus b}. \tag{2}$$

There, $\Phi_f$ is the cumulant-generating function of the root law distribution $f$: $\Phi_f(\theta) \triangleq \log \int_{\mathfrak{R}} e^{r\theta} df(r)$. As soon as $f$ has finite exponential moments, $\Phi_f$ is well-defined and convex; in particular, (2) is a strongly convex problem with a unique minimizer [10].

We recall below Theorem 2 [10], that guarantees monotonicity of $\mathrm{GBT}_{f,\sigma}$, when two elements can only be compared once. The forthcoming Theorem 3 extends the result to situations where two elements are compared multiple times.

**Proposition 1** (Th. 2, [10]). *Consider a root law $f$, a scalar $\sigma > 0$, and two datasets $\mathbf{D}$, $\mathbf{D}'$ which contains at most one comparison between any pair $(a, b) \in \mathcal{A}^2$. Then, for all $a$, $\mathbf{D} \succeq_a \mathbf{D}'$ implies $\mathrm{GBT}_{f,\sigma}(\mathbf{D}) \geq_a \mathrm{GBT}_{f,\sigma}(\mathbf{D}')$.*

Although well behaved in many aspects, the generalized Bradley-Terry model fails to perform generalization: an alternative $a$ that never appears in $\mathbf{D}$ will receive a nil score $\mathrm{GBT}(\mathbf{D})_a = 0$. However, in practice, alternatives may *(i)* admit informative descriptions, and *(ii)* have known relationships. This should help guess the score of a yet uncompared alternative, based on the scoring of similar compared alternatives. We introduce such a learning algorithm in the next Section.

---

[2]Formally, ALG$(\sigma \cdot \mathbf{D}) = \sigma \cdot$ ALG for all permutations of $\mathcal{A}$, with the action that applies pointwise to all apparitions of an alternative $a \in \mathcal{A}$.

[3]In [10], the authors consider datasets that contain at most one comparison per pair of alternatives.

# 3 Linear GBT with Diffusion Prior

In this Section, we introduce a class of preference learning algorithms that incorporate both user comparisons and contextual information on the compared elements, and state our main result about their mathematical guarantees on monotonicity.

## 3.1 Learning with prior similarities

The $\text{GBT}_{\sigma,f}$ model does not include prior knowledge on the structure of the alternatives. For example, when alternatives represent videos on YouTube, the fact that two videos belong to the same channel cannot be represented in Equation (2). More generally, Equation (2) does not encode any prior similarities between alternatives. Consequently, if an alternative $a$ is never compared with any other, then, even if $a$ is similar to an alternative $b$ that has a large non-zero score, $a$ will still be assigned a zero score. In other words, Equation (2) does not allow us to generalize.

To address this issue, we generalize $\text{GBT}_{\sigma,f}$ (2) in two directions.

1. *(Embeddings)* We assume that, to each alternative $a \in \mathcal{A}$, corresponds an *embedding* $x_a \in \mathbb{R}^D$, where $D$ is a positive integer. We model the score of alternative $a$ by a linear function of the embedding:
$$\theta_a(\beta) \triangleq x_a^T \beta,$$
for a parameter $\beta$. Denoting $x \in \mathbb{R}^{D \times A}$ the matrix collecting all embeddings, and $x_{a \ominus b} = x_a - x_b$ for any $a, b \in \mathcal{A}$, the GBT parameter $\theta \in \mathbb{R}^{\mathcal{A}}$ is replaced by a linear function $\theta(\beta) = x^T \beta$. For instance, in the context of YouTube, $x_a$ could denote a one-hot encoding the content creator identity; more in Section 5.

2. *(Similarity)* We consider a more general regularization term $\mathcal{R}(\beta)$ of the form
$$\mathcal{R}(\beta) = \frac{1}{2\sigma^2} \sum_d \beta_d^2 + \frac{1}{2} \sum_{ab} \theta_a(\beta) L_{ab} \theta_b(\beta) \tag{3}$$

where $L$ is a Laplacian matrix, i.e. such that $L_{aa} = \sum_{b \neq a} |L_{ab}| \geq 0$ and $L_{ab} = L_{ba} \leq 0$, for all $a \neq b$. The matrix $L$ can be thought as the Laplacian of a graph encoding (prior) similarities between alternatives, the weight $|L_{ab}|$ representing the similarity between $a$ and $b$. Therefore, the regularization term $\sum_{ab} \theta_a(\beta) L_{ab} \theta_b(\beta) = \frac{1}{2} \sum_{a \neq b} |L_{ab}| (\theta_a(\beta) - \theta_b(\beta))^2$ incentivizes the model to (a priori) assign similar scores to similar alternatives.

We can now define the class of GBT models that we will study in this paper.

**Definition 3** (Linear GBT with Diffusion Prior). *Let $f$ be a root law, $x$ be an embedding, $\sigma > 0$ a positive constant, and $L$ a Laplacian matrix. The model $\text{GBT}_{f,\sigma,x,L}$ is defined as*
$$\text{GBT}_{f,\sigma,x,L}(\mathbf{D}) \triangleq x^T \beta^*(\mathbf{D}) \in \mathbb{R}^A,$$

*where $\beta^*(\mathbf{D}) \triangleq \arg\min \mathcal{L}(\cdot|\mathbf{D})$ minimizes the strongly convex loss function*
$$\mathcal{L}(\beta|\mathbf{D}) = \mathcal{R}(\beta) + \sum_{(a,b,r) \in \mathbf{D}} \Phi_f(x_{a \ominus b}^T \beta) - r x_{a \ominus b}^T \beta. \tag{4}$$

*For conciseness, let $\theta^*(\mathbf{D}) \triangleq \text{GBT}_{f,\sigma,x,L}(\mathbf{D}) = x^T \beta^*(\mathbf{D})$.*

Remark that the original GBT is a special case of Linear GBT with Diffusion Prior with $A = D$, $x = I$ the identity matrix, and $L = 0$.

**Proposition 2.** *Linear GBT with diffusion prior is* neutral, *i.e. invariant up to alternative relabeling.*

*Proof.* See Appendix B for a formal statement and derivation. $\square$

## 3.2 Monotonicity and diffusion

We now present our main result (Theorem 1). We prove that for a special class of embeddings, namely *diffusion embeddings*, monotonicity is guaranteed. Diffusion embeddings take their name from the interplay with (super) laplacian matrices.

**Definition 4** (Super-Laplacian matrix). *A super-Laplacian matrix $\Delta$ is a symmetric matrix such that for all $a \neq b$, $\Delta_{aa} > -\sum_{b \neq a} \Delta_{ab}$ and $\Delta_{ab} \leq 0$.*

**Definition 5** (Diffusion embedding). *An embedding $x$ is a diffusion embedding if the Gram matrices $X_\lambda = x^T x + \lambda I$ have super-Laplacian inverses $X_\lambda^{-1}$ for any $\lambda > 0$.*

Note that if $X = x^T x$ is itself invertible with super-Laplacian inverse, then it is a diffusion embedding. However, this case is restrictive since it implies that $D \geq A$.

**Theorem 1** (Monotonicity with diffusion embeddings). *For any root law $f$, positive constant $\sigma > 0$, diffusion embedding $x$, and Laplacian matrix $L$, $\text{GBT}_{f,\sigma,x,L}$ is monotone.*

*Proof.* The theorem follows directly from Proposition 3 and Theorem 3, which are provided and proved in Section 4. $\square$

### 3.3 Example: one-hot encoding

A one-hot encoding is possible when the alternatives can be arranged into multiple disjoint classes. For example, if the alternatives represent videos on YouTube, one can partition them by the YouTube channel they belong to. In that case, the score of an alternative $a$ is defined as $\theta_a = \gamma_{d(a)} + s^2 \cdot \alpha_a$, where $d(a)$ is the channel of $a$. The score $\gamma_{d(a)}$ represents the score of the channel $d(a)$, while $\alpha_a$ represents a residual score of $a$, and $s$ is a real constant that controls the scale of the residual score. Theorem 2 states a one-hot encoding is an example of diffusion embedding. We postpone the proof to Appendix H.

**Theorem 2** (GBT with one-hot encoding). *Let $f$ be a root law, $\sigma > 0$ a positive constant, $L$ a Laplacian matrix and $s \in \mathbb{R}$. Let $x : \mathbb{R}^{D \times A}$ be a one-hot encoding matrix: $x_{da} = 1$ if, and only if, $a$ belongs to $d$. Then, for any real number $s$, $(x \quad sI)^T$ is a diffusion embedding and the score $\text{GBT}_{f,\sigma,x,L}$ is monotone.*

## 4 The proof

This Section provides the mathematical analysis that builds to the proof of the main result, Theorem 1. Section 4.1 proposes a differential analysis framework for the dataset operations outlined above; Section 4.2 then provides the proof.

### 4.1 Differential analysis of dataset operations

The goal of this technical section is to connect the discrete domain of datasets with tools from differential analysis. Studying the monotonicity (Definition 2) of $\theta^*(\mathbf{D})$ requires to compare the loss functions for datasets that are related by a basic operation. Given that the loss is invariant under EXCHANGE and SHUFFLE operations on the dataset $\mathbf{D}$, on one hand because of the specific form of the GBT loss, and on the other because it features a sum of comparison samples of the dataset, the same invariance holds for $\theta^*$. Thus, to prove monotonicity, it suffices to study what happens under APPEND and UPDATE operations that favor $a$ over $b$.

Because the loss function is a sum of terms indexed by the elements of the dataset, this relation is quite simple

$$\mathcal{L}(\beta| \text{APPEND}_{a,b,r}(\mathbf{D})) = \mathcal{L}(\beta|\mathbf{D}) + \Phi_f(\theta_{a \ominus b}(\beta)) - r\theta_{a \ominus b}(\beta) \tag{5}$$

$$\mathcal{L}(\beta| \text{UPDATE}_{n,r}(\mathbf{D})) = \mathcal{L}(\beta|\mathbf{D}) - (r - r_n)\theta_{a_n \ominus b_n}(\beta) \tag{6}$$

To enable the differential analysis of these operations, we introduce a smooth deformation of the loss function.

**Definition 6** (Smoothed loss). *For every $\lambda \in \mathbb{R}$, and every operation $o$ of the form $\text{APPEND}_{a,b,r}$ or $\text{UPDATE}_{n,r}$, we define the smoothed loss $\mathcal{L}_\lambda$ by*

$$\mathcal{L}_\lambda(\beta|\mathbf{D}, o) \triangleq \mathcal{L}(\beta|\mathbf{D}) + \lambda \cdot \begin{cases} \Phi_f(\theta_{a \ominus b}(\beta)) - r\theta_{a \ominus b}(\beta) & \textit{if } o = \text{APPEND}_{a,b,r}, \\ -(r - r_n) \cdot \theta_{a_n \ominus b_n}(\beta) & \textit{if } o = \text{UPDATE}_{n,r}, \\ 0 & \textit{otherwise.} \end{cases} \tag{7}$$

*Denote also $\beta_\lambda^*(\mathbf{D}, o) \triangleq \arg\min \mathcal{L}_\lambda(\cdot|\mathbf{D}, o)$ and $\theta_\lambda^*(\mathbf{D}, o) \triangleq \theta(\beta_\lambda^*(\mathbf{D}, o))$.*

The smoothed loss matches the loss at $\lambda \in \{0, 1\}$, as $\mathcal{L}_0(\beta|\mathbf{D}, o) = \mathcal{L}(\beta|\mathbf{D})$ and $\mathcal{L}_1(\beta|\mathbf{D}, o) = \mathcal{L}(\beta|o(\mathbf{D}))$. We will leverage this by using the integral expression

$$\theta^*(o(\mathbf{D})) - \theta^*(\mathbf{D}) = \int_0^1 \frac{d\theta^*_\lambda}{d\lambda}(\mathbf{D}, o) d\lambda. \tag{8}$$

When $\lambda \mapsto \theta^*_\lambda(\mathbf{D}, o)$ is continuously differentiable, this integral expression is well-defined, and it suffices that the derivative $d\theta^*_{\lambda a}(\mathbf{D}, o)/d\lambda$ be non-negative for the score difference at $a$ to be non-negative. Lemma 1 states that this derivative is well defined and provides a formula. The proof is given in Appendix C.

**Lemma 1.** *Let $H = H(\theta|\mathbf{D})$ denote the Hessian of $\mathcal{E}(\theta|\mathbf{D}) = \sum_{(a,b,r) \in \mathbf{D}} \Phi_f(\theta_{a \ominus b})$, and $X = \sigma^2 x^T x$ denote the (scaled) Gram matrix of the embedding $x$. Then, for any basic operation $o$ and dataset $\mathbf{D}$, the loss function $\mathcal{L}_\lambda(\cdot|\mathbf{D}, o)$ admits a unique global minimizer $\beta^*_\lambda(\mathbf{D}, o)$, and the inferred score $\theta^*_\lambda(\mathbf{D}, o)$ is a smooth function of $\lambda$ over $[0, \infty)$. Moreover,*

- *if $o = \text{UPDATE}_{n,r}$ and $\mathbf{D}_n \simeq (a, b, s)$, then the score of the alternative $a$ satisfies*

$$\left.\frac{d\theta^*_{\lambda a}}{d\lambda}(\mathbf{D}, o)\right|_{\lambda=\mu} = (r - s) \cdot e_a^T \left(I + X(L + H)\right)^{-1} X e_{a \ominus b}. \tag{9}$$

  *There, $e_a$ denotes the $a$-th vector of the cartesian basis of $\mathbb{R}^A$, $\theta^*_{\lambda a}$ denotes the $a$-th coordinate of $\theta^*_\lambda$, and $e_{a \ominus b} = e_a - e_b$.*
- *if $o = \text{APPEND}_{a,b,r}$, then the score of the alternative $a$ satisfies*

$$\left.\frac{d\theta^*_{\lambda a}(\mathbf{D}, o)}{d\lambda}\right|_{\lambda=\mu} = (r - \Phi'_f(\theta^*_{a \ominus b})) \cdot e_a^T \left(I + X(L + H + \mu \Phi''_f(\theta_{a \ominus b}) \cdot S^{ab})\right)^{-1} X e_{a \ominus b}, \tag{10}$$

  *where $S^{ab} \in \mathbb{R}^{A \times A}$ is the Laplacian matrix of the graph over the alternatives with a single edge $ab$ (with weight 1), i.e. $S^{ab}_{aa} = S^{ab}_{bb} = 1$, $S^{ab}_{ab} = S^{ab}_{ba} = -1$, and $S^{ab}_{cd} = 0$ otherwise.*

We are interested in the sign (positive or negative) of the expressions in Equations (9) and (10). First, the factors $r - s$ and $r - \Phi'_f(\theta^*_{a \ominus b})$ are easy to understand. If $o = \text{UPDATE}_{n,r}$ favors $a$ over $b$ then $r - s \geq 0$ by definition. If $o = \text{APPEND}_{a,b,r}$ favors $a$ over $b$, then $r = \sup \mathfrak{R}$ which is the supremum of $\Phi'_f$ [10, Theorem 1].

Therefore, if we want to compare the scores of $a$ and $b$, the important factor to study is the matrix $(I + X(L + \tilde{H}))^{-1} X$, where $\tilde{H} = H + \mu \Phi''_f(\theta_{a \ominus b}) \cdot S^{ab}$ if $o = \text{APPEND}_{a,b,r}$, or $\tilde{H} = H$ if $o = \text{UPDATE}_{n,r}$ and $\mathbf{D}_n \simeq (a, b, s)$. We study this matrix in the next section.

### 4.2 Good embeddings: a sufficient condition for monotonicity

In this Section, we provide a sufficient condition on the embedding for the Linear GBT model to be monotone (Definition 7, Theorem 3), and provide mathematical properties of this condition (Proposition 9). Finally, we show that diffusion embeddings meet the above sufficient condition (Proposition 3).

**Definition 7** (Good embeddings). *Given a Laplacian matrix $Y$, an embedding $x$ is $Y$-good if the Gram matrix $X = x^T x$ satisfies $e_a^T (I + XY)^{-1} X e_{a \ominus b} \geq 0$ for all $(ab)$. An embedding $x$ is good if $x$ is $Y$-good for all Laplacian matrices $Y$.*

**Theorem 3** (Monotonicity with good embeddings). *For any root law $f$, positive constant $\sigma > 0$, Laplacian matrix $L$, and good embedding $x$, $\text{GBT}_{f,\sigma,x,L}$ is monotone.*

Before going to the proof, we provide some intuition. Note first that the Hessian $H$ of $\mathcal{E}(\theta|\mathbf{D}) = \sum_{(a,b,r) \in \mathbf{D}} \Phi_f(\theta_{a \ominus b})$ is also a Laplacian matrix. Indeed, let $G$ be the weighted graph whose edges are the pairs $(ab)$ of alternatives that occur in the dataset $\mathbf{D}$, weighted by $G_{ab} = N_{ab} \cdot \Phi''_f(\theta_{a \ominus b})$ where $N_{ab}$ is the number of occurrences of the pair $(ab)$ in $\mathbf{D}$. Since $\Phi_f$ is convex, these weights are nonnegative. Then, a direct calculation shows that $H$ is the graph Laplacian of the weighted graph $G$: for $a \neq b$, $H_{aa} = \sum_{c \neq a} N_{ac} \Phi''_f(\theta_{ac})$ and $H_{ab} = -N_{ab} \Phi''_f(\theta_{a \ominus b})$. Therefore, given any prior Laplacian matrix $L$, the matrix $L + \tilde{H}$, where $\tilde{H}$ is defined in section 4.1, is the Laplacian of a graph that combines the prior similarities ($L$) with the similarities inferred from the dataset at hand ($\tilde{H}$). This observation motivates Definition 7.

*Proof of Theorem 3.* By Lemma 1, if, for every operation $o$, every score function $\theta$ and every dataset $\mathbf{D}$, the inequality $e_a^T(I + X(L + \tilde{H}))^{-1}Xe_{a\ominus b} \geq 0$ holds, then the score $\theta^*(\mathbf{D})$ derived from the loss function of Equation (4) is monotone. This is precisely implied by $x$ being good. □

This result motivates a more precise understanding of good embeddings. In the special cases $(A, D) = (2, D)$ and $(A, D) = (A, 1)$, we have complete characterizations (see Appendix D). In general, however, checking goodness is not straightforward: two embeddings $x$ and $y$ can be individually good, while their concatenation $\begin{bmatrix} x & y \end{bmatrix}^\top$ fails to be good (see Propositions 7 and 8, Appendix E). Nonetheless, any embedding can be made $Y$-good by concatenating it with a sufficiently scaled identity. We formalize this in Appendix F.

### 4.3 Diffusion embeddings are good

Finally, we show that any diffusion embedding is a good embedding. This uses the fact that any super-Laplacian matrix $\Delta$ satisfies $e_a^T\Delta^{-1}e_{a\ominus b} \geq 0$ for any pair $(a, b) \in \mathcal{A}^2$. This result has been proved in [10, Lemma 1] and we provide an alternative proof highlighting the diffusion perspective G.

**Proposition 3.** *Any diffusion embedding is a good embedding.*

*Proof.* If $x$ is a diffusion embedding, $\lambda > 0$, and $Y$ is an arbitrary Laplacian matrix, then the matrix $X_\lambda^{-1} + Y$, where $X_\lambda = x^Tx + \lambda I$, is super-Laplacian. Consequently, $e_a^T(I + X_\lambda Y)^{-1}X_\lambda e_b = e_a^T(X_\lambda^{-1} + Y)^{-1}e_b \geq 0$. The claim follows by taking the limit $\lambda \to 0$. □

## 5 Experimental evaluation

In this Section, we provide a numerical exploration of the prevalence of "goodness" for random embeddings, and the statistical error of several preference learning models.[4] Appendix A provides complementary experiments on real-world data.

### 5.1 Probability of goodness for random embeddings

To illustrate the challenges of achieving good embeddings, we generate random i.i.d. Gaussian embedding matrices x and evaluate their quality. In Figure 1, we examine a single Gaussian embedding $x$ (left) and its concatenation with the identity matrix $I$ (right). Our findings indicate that the goodness of $x$ is more likely for large values of $D/A$ and significantly diminishes when $A/D$ is large. The concatenation with the identity matrix notably enhances the goodness, aligning with Proposition 9.

### 5.2 Generative model, metric, and simulations

For each experiment, we consider the ground-truth embedding $x^\dagger \in \mathbb{R}^{D \times A}$, Laplacian matrix $L^\dagger$, constant $\sigma^\dagger$, and root law $f^\dagger$. The ground-truth features are generated as $\beta^\dagger \sim \mathcal{N}(0, (\sigma^\dagger)^2I + x^\dagger L^\dagger(x^\dagger)^T)$ and $\theta^\dagger = (x^\dagger)^T\beta^\dagger$. We then create a dataset $\mathbf{D} : [N] \to \mathcal{A}^2 \times \mathfrak{R}$ by first selecting $N$ random comparison pairs uniformly. The corresponding random comparisons $r$ are generated using the root law $f^\dagger$ and conditionally to $\theta^\dagger$. We shall only consider the uniform root law $f^\dagger = \frac{1}{2}1_{[-1,1]}$ and set $\sigma^\dagger = 1$.

The estimated scores are computed as $\theta^*(\mathbf{D}) = \text{GBT}_{f,\sigma,x,L}(\mathbf{D})$, where the quadruplet $(f, \sigma, x, L)$ may or may not align with the ground truth. Since the quality of a score vector is invariant under constant shifts, we evaluate the error over zero-mean versions of both $\theta^\dagger$ and $\theta^*(\mathbf{D})$. More precisely, we use Monte Carlo simulations to estimate the normalized mean squared error (nMSE), defined as:

$$\text{nMSE}(f^\dagger, \sigma^\dagger, x^\dagger, L^\dagger; f, \sigma, x, L; N) = \text{nMSE} = \mathbb{E}\left[\frac{\left\|\left(\theta^*(\mathbf{D}) - \bar{\theta}^*(\mathbf{D})\right) - \left(\theta^\dagger - \bar{\theta}^\dagger\right)\right\|^2}{\|\theta^\dagger - \bar{\theta}^\dagger\|^2}\right].$$

We then analyze how the nMSE evolves with respect to various parameters.

---

[4]The code is available at `https://github.com/pevab/gbtlab2`, and will be made publicly after the review process. We run experiments on a personal laptop with 16GB of RAM and a 2.10 GHz processor.

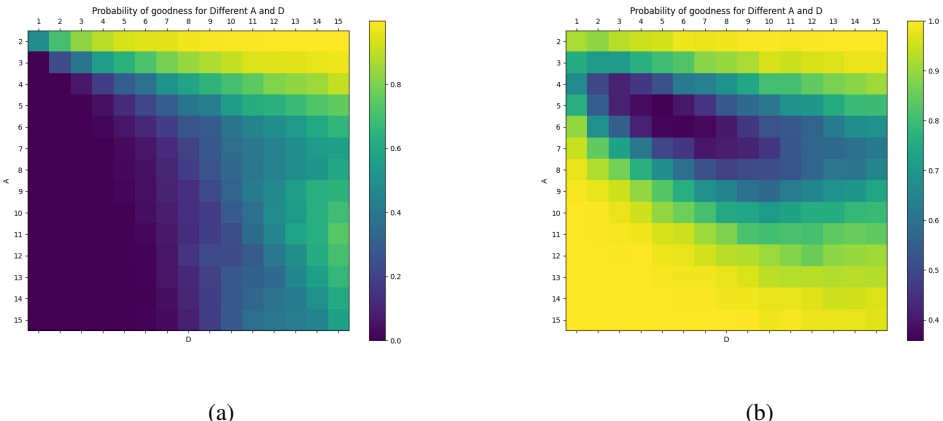

(a)                                   (b)

Figure 1: *Left pane:* Probability that a Gaussian i.i.d embedding $x$ is a good embedding for $2 \leq A \leq 15$ and $1 \leq D \leq 15$. *Right pane:* As for the left pane with embedding $\begin{bmatrix} I & x \end{bmatrix}^T$.

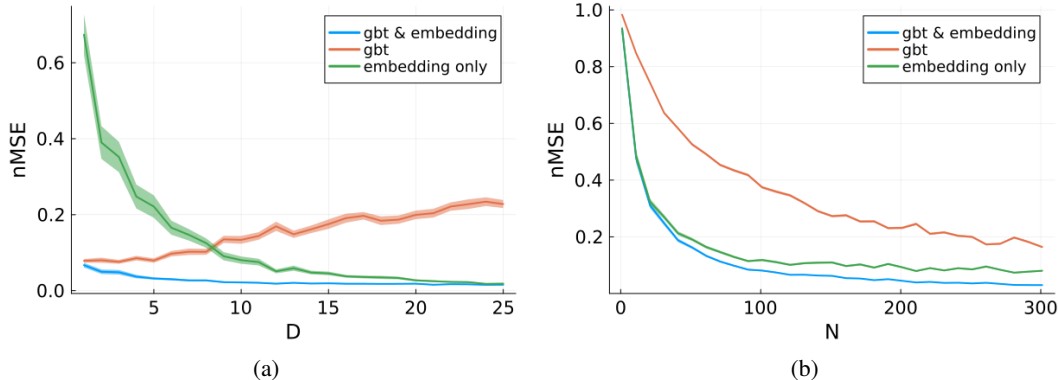

(a)                                   (b)

Figure 2: *Left pane:* nMSE as a function of $D$ for $A = 25$ alternatives and $N = 500$ comparisons over 100 seeds. Blue curve with $\begin{bmatrix} I & x \end{bmatrix}^T$ (full embedding), orange curve with embedding $I$ (classical GBT), and green curve with embedding $x$ (features only). *Right pane:* nMSE with respect to the number of comparisons $N$ for $A = 20$, $D = 10$, and 1000 seeds. Blue curve: GBT with one-hot encoding; Orange curve: GBT. Every curve is displayed with its error bar (using $1.96\sigma/\sqrt{\text{n\_seeds}}$).

Figure 2a shows the nMSE as a function of $D$, using data generated with $x^\dagger = \begin{bmatrix} I & \tilde{x}^\dagger \end{bmatrix}^T$ (i.i.d. Gaussian $\tilde{x}$), uniform $f^\dagger$, $L^\dagger = 0$, and $\sigma^\dagger = 1$. We compare three models with shared parameters $(f, L, \sigma) = (f^\dagger, L^\dagger, \sigma^\dagger)$, using $x^\dagger$, $I$ (classical GBT), and $\tilde{x}^\dagger$ respectively. The embedding-based model outperforms others, combining the strengths of classical GBT for small $D$ and feature-based learning for larger $D$.

Figure 2b shows the nMSE as a function of the number of comparisons $N$. Data are generated with $(f^\dagger, x^\dagger, L^\dagger, \sigma^\dagger) = \left( \frac{1}{2} 1_{[-1,1]}, \begin{bmatrix} I & \tilde{x}^\dagger \end{bmatrix}^T, 0, 1 \right)$, where $\tilde{x}$ is a one-hot encoding matrix (see Section 3.3). The results show that one-hot encoding greatly reduces the number of comparisons needed to reach a given nMSE. This is useful in applications like YouTube score estimation [17], where the encoding reflects the channel and enables generalization across alternatives.

## 6 Conclusion

In this paper, we introduced a new comparison-based preference learning model, namely *linear GBT with diffusion prior*. This model not only generalizes to previously uncompared data using embeddings, but also potentially guarantees monotonicity, depending on the class of embeddings used. We proved that our model is monotone for various classes of embeddings (one-hot encodings,

diffusion, and good embeddings). To the best of our knowledge, linear GBT with diffusion prior is the first model that guarantees monotonicity while being able to generalize.

Diffusion embeddings form a class of embeddings (containing one-hot encodings) that yield monotonicity. Our proof techniques relied on an interesting interplay between an algebraic criterion for monotonicity (Definition 7) and properties of (super) Laplacian matrices akin to diffusion theory.

**Limitations.** While improving the understanding of preference learning with guarantees, our theory currently provides guarantees for diffusion embeddings only. We hope to motivate more work on preference learning with guarantees, in order to build more truseworthy AI systems, with notable applications in [39, 16, 17, 34]. Also, we caution against the use of preference learning algorithms that rely on data collected in inhumane conditions, as is mostly the case today [18, 29, 15, 14]. It is unclear whether our work can positively contribute to this issue.

### Acknowledgements

The contribution of Gilles Bareilles has been funded by European Union's Horizon Europe research and innovation programme under grant agreement No. 101070568.

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

# A Experiments on real world data

In this Section, we complete the experiments on synthetic data (Section 5) with experiments on real-world data [16]. More precisely, we provide numerical evidence for the fact that including a (diffusion) embeddings improves performance. The code to reproduce experiments is available at `https://github.com/pevab/gbtlab2`.

## A.1 Experimental set-up

The real-world data contains comparisons between Youtube videos made by various users, from the Tournesol platform [17]. We selected a subset $\mathbf{D}$ of 1000 comparisons from a single user, the one that has most comparisons. Every comparison is a tuple $(a, b, r)$ where $a, b$ are video identifiers, and $r$ is the comparison value $r$, originally an integer between $-10$ and $10$, which we rescale to fit in $[-1, 1]$.

In addition, we associate for each video $a$, the YouTube channel $c$ it belongs to. We describe this relation using a one-hot encoding matrix $\chi \in \mathbb{R}^{D \times N}$: $\chi_{ca} = 1$ if the video $a$ belongs to the channel $c$, or $\chi_{ca} = 0$ otherwise. We obtain an embedding $x \in \mathbb{R}^{(D+N) \times N}$ by concatenating $\chi$ with $\lambda I$, that is, $x = \begin{pmatrix} \chi \\ I_N \end{pmatrix}$.

We compare two models: *(i)* $\text{GBT}_{f,1,x,0}$ (which uses embeddings), and *(ii)* $\text{GBT}_{f,1,I_N,0}$ (the original GBT, which does not use embeddings). Both models have the uniform distribution in $[-1, 1]$ as a root law, $f(r) = \frac{1}{2}1_{[-1,1]}(r)$, and the same Gaussian prior. We do not use any Laplacian regularization.

After training, each model $M$ computes, given a pair $(a, b)$ of video identifiers, the expected comparison value defined as

$$M(a, b) = \int r \cdot f(r) e^{r(\theta_a^* - \theta_b^*)} dr = \Phi_f'(\theta_a^* - \theta_b^*) \tag{11}$$

where $\Phi_f$ is the cumulant-generating function of the root law, and $\theta^*$ are the scores learned. Given a validation dataset $\mathbf{D}_{\text{val}}$, the validation risk of $M$ is given by

$$\frac{1}{|\mathbf{D}_{\text{val}}|} \sum_{(a,b,r) \in \mathbf{D}_{\text{val}}} (M(a, b) - r)^2 \tag{12}$$

## A.2 Results

Figure 3 reports the empirical risks of the two models, using a 10-fold cross validation scheme over the dataset $\mathbf{D}$ (1000 comparisons). We observe that the average validation risk of the model with embeddings is $8.40 \cdot 10^{-3}$, while that of the model without embeddings is $10.1 \cdot 10^{-3}$. Hence, including the YouTube channel embeddings reduces the risk by $17\%$ on average.

# B Proof of neutrality (Proposition 2)

To formalize neutrality, we must define how a permutation $\tau$ of the alternatives acts on the inputs and outputs of Linear GBT with Diffusion Prior. We define the actions as follow:

$$(\tau \cdot \mathbf{D})_n = \tau \cdot \mathbf{D}_n, \tag{13}$$
$$\tau \cdot (a, b, r) = (\tau(a), \tau(b), r), \tag{14}$$
$$(\tau \cdot \theta)_a = \theta_{\tau(a)}, \tag{15}$$
$$(\tau \cdot x)_a = x_{\tau(a)}, \tag{16}$$
$$(\tau \cdot L)_{ab} = L_{\tau(a)\tau(b)}. \tag{17}$$

Neutrality is formalized by the equality

$$\forall \tau, \ \tau^{-1} \cdot GBT_{f,\sigma,\tau \cdot x, \tau \cdot L} \circ \tau = GBT_{f,\sigma,x,L}. \tag{18}$$

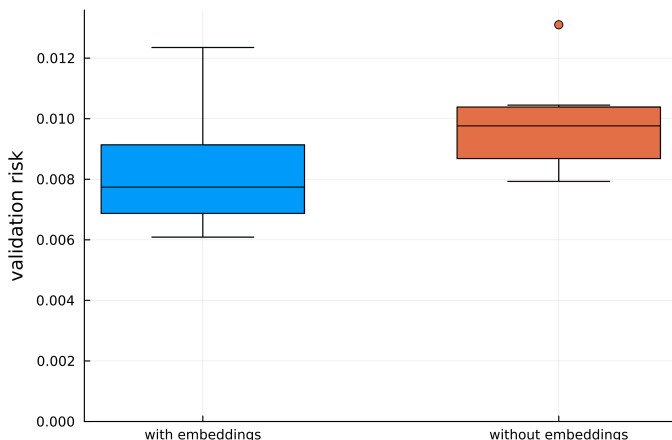

Figure 3: Validation risks of $\text{GBT}_{f,1,x,0}$ (with embeddings, on the left) and $\text{GBT}_{f,1,I_N,0}$ (without embeddings, on the right), estimated using 10-fold cross validation. The box plots report the minimum, 1st quartile, median, 3rd quartile and maximum. Outliers are also shown.

We indeed have

$$\left(\tau^{-1} \cdot GBT_{f,\sigma,\tau\cdot x,\tau\cdot L} \circ \tau(\mathbf{D})\right)_a = \left(GBT_{f,\sigma,\tau\cdot x,\tau\cdot L}\left(\tau\cdot\mathbf{D}\right)\right)_{\tau^{-1}(a)} \tag{19}$$

$$= (\tau\cdot x)_{\tau^{-1}(a)}^T \beta_{f,\sigma,\tau\cdot x,\tau\cdot L}^*(\tau\cdot\mathbf{D}) = x_a^T \beta_{f,\sigma,\tau\cdot x,\tau\cdot L}^*(\tau\cdot\mathbf{D}) \tag{20}$$

$$= x_a^T \arg\min_\beta \frac{1}{2\sigma^2}\|\beta\|_2^2 + \frac{1}{2}\sum_{ab}(x_{\tau(a)}^T\beta)L_{\tau(a)\tau(b)}(x_{\tau(b)}^T\beta)$$

$$\qquad + \sum_{(a,b,r)\in\mathbf{D}}\left(\Phi_f((\tau\cdot x)_{\tau^{-1}(a)\tau^{-1}(b)}^T\beta) - r(\tau\cdot x)_{\tau^{-1}(a)\tau^{-1}(b)}^T\beta\right) \tag{21}$$

$$= x_a^T \arg\min_\beta \frac{1}{2\sigma^2}\|\beta\|_2^2 + \frac{1}{2}\sum_{a'b'}(x_{a'}^T\beta)L_{a'b'}(x_{b'}^T\beta) + \sum_{(a,b,r)\in\mathbf{D}}\left(\Phi_f(x_{a\ominus b}^T\beta) - rx_{a\ominus b}^T\beta\right) \tag{22}$$

$$= x_a^T \beta_{f,\sigma,x,L}^*(\mathbf{D}) = \left(GBT_{f,\sigma,x,L}(\mathbf{D})\right)_a. \tag{23}$$

## C  Proof of Lemma 1

We consider the case of $o = \text{APPEND}_{a,b,r}$. The case $o = \text{UPDATE}_{n,r}$ with $\mathbf{D}_n \simeq (a,b,s)$ is proved similarly. The loss functions $\mathcal{L}_\lambda(\beta|\mathbf{D},o)$ and $\mathcal{L}(\beta|\mathbf{D})$ differ by $\lambda(\Phi_f(\theta_{a\ominus b}(\beta)) - r\theta_{a\ominus b}(\beta))$. The term $\theta_{a\ominus b}$ being linear in $\beta$, its Hessian is zero. On the other hand, the Hessian of $\Phi_f(\theta_{a\ominus b})$ is simply the Laplacian $\Phi_f''(\theta_{a\ominus b}) \cdot S^{ab}$ of the graph with a single edge $ab$ with weight $\Phi_f''(\theta_{a\ominus b})$. Writing

$$H^\lambda = H + \lambda\Phi_f''(\theta_{a\ominus b}) \cdot S^{ab}, \tag{24}$$

we obtain the Hessian of the loss

$$H_\beta\mathcal{L}_\lambda(\beta|\mathbf{D},o) = x(L + H^\lambda)x^T + \frac{1}{\sigma^2}I \succeq \frac{1}{\sigma^2}I. \tag{25}$$

Therefore, the loss $\mathcal{L}_\lambda(\beta|\mathbf{D},o)$ is strictly convex, and admits a global minimizer $\beta_\lambda^*(\mathbf{D},o)$. To simplify notations in this proof, we write $\beta^*(\lambda)$.

We want to use the implicit function theorem to analyze how $\beta^*(\lambda)$ varies with $\lambda$. For that, consider the gradient of the loss $\mathcal{L}_\lambda(\beta|\mathbf{D},o)$

$$F(\beta,\lambda) = \nabla_\beta\mathcal{L}(\beta|o_\lambda(\mathbf{D})) = \nabla_\beta\mathcal{L}(\beta|\mathbf{D}) + \lambda\cdot(\Phi_f'(\theta_{a\ominus b}) - r)xe_{a\ominus b} \tag{26}$$

with domain $\mathbb{R}^D \times \mathbb{R}$ and codomain $\mathbb{R}^D$.

Fix some $\mu \in \mathbb{R}$. The Jacobian of $F(\beta,\lambda)$ with respect to $\beta$, evaluated at $\lambda = \mu$, is given by

$$J_\beta F(\beta,\mu) = \frac{1}{\sigma^2}I + x(L + H^\mu)x^T \tag{27}$$

which is invertible. Moreover, since $\beta^*(\mu)$ is a minimizer, we have $F(\beta^*(\mu), \mu) = 0$.

Hence, the implicit function theorem states that there exists an open neighborhood $U$ of $\mu$, and a smooth function $\gamma : U \to \mathbb{R}^D$ such that

$$\gamma(\mu) = \beta^*(\mu) \tag{28}$$

$$\forall \lambda \in U, F(\gamma(\lambda), \lambda) = 0 \tag{29}$$

The latter equality implies that $\gamma(\lambda) = \beta^*(\lambda)$ for all $\lambda \in U$. Consequently, $\beta^*(\lambda)$ and $\theta^*(\lambda) = x^T \beta^*(\lambda)$ depend smoothly on $\lambda$. In addition, the implicit function theorem also gives an expression for the Jacobian $J_\lambda \beta^*$, evaluated at $\mu$

$$J_\lambda \beta^*(\mu) = -(J_\beta F)^{-1} J_\lambda F \tag{30}$$

$$= (r - \Phi'_f(\theta^*_{a \ominus b})) \cdot \left( \frac{1}{\sigma^2} I + x(L + H^\mu) x^T \right)^{-1} x e_{a \ominus b} \tag{31}$$

Finally, note that

$$\frac{d\theta^*}{d\lambda}(\mu) = J_\beta \theta^*(\beta^*(\mu)) \cdot J_\lambda \beta^*(\mu) \tag{32}$$

$$= (r - \Phi'_f(\theta^*_{a \ominus b})) \cdot x^T \left( \frac{1}{\sigma^2} I + x(L + H^\mu) x^T \right)^{-1} x e_{a \ominus b} \tag{33}$$

Let $M = L + H^\mu$, and $X = \sigma^2 x^T x$. Using Woodbury's identity $(I + UV)^{-1} = I - U(I + VU)^{-1} V$, we derive

$$\left( \frac{1}{\sigma^2} I + x(L + H^\mu) x^T \right)^{-1} = \sigma^2 (I + \sigma^2 x M x^T)^{-1} \tag{34}$$

$$= \sigma^2 I - \sigma^4 x M (I + \sigma^2 x^T x M)^{-1} x^T \tag{35}$$

$$x^T \left( \frac{1}{\sigma^2} I + x(L + H) x^T \right)^{-1} x = X - X M (I + X M)^{-1} X \tag{36}$$

$$= (I - X M (I + X M)^{-1}) X \tag{37}$$

$$= (I + X M)^{-1} X \tag{38}$$

Thus,

$$\frac{d\theta^*}{d\lambda}(\mu) = (r - \Phi'_f(\theta^*_{a \ominus b})) \cdot (I + \sigma^2 x^\top x(L + H^\mu)) \sigma^2 x^\top x e_{a \ominus b}. \tag{39}$$

The result follows.

## D   Good Embeddings for $A = 2$ or $D = 1$

We can fully characterize goodness in lowest dimensional regime, either with only two alternatives or with an embedding on a single feature.

**Definition 8.** *We say that a matrix $M$ is max-diagonally dominant if $M_{aa} \geq M_{ab}$ for any $(ab)$.*

**Proposition 4.** *An embedding with Gram matrix $X = \begin{bmatrix} a & c \\ c & b \end{bmatrix}$ is monotonicity proof if and only if*

$-\sqrt{ab} \leq c \leq \min(a, b)$.

*Proof.* We first observe that, since $x^T x$ is positive semidefinite, $a, b \geq 0$ and $det(x^T x) = ab - c^2 \geq 0$. This implies in particular that $c \geq -\sqrt{ab}$, which is the desired lower bound.

Two-dimensional Laplacian matrices are of the form $Y = \delta \begin{bmatrix} 1 & -1 \\ -1 & 1 \end{bmatrix}$ with $\delta > 0$, which can be chosen as being equal to 1 without loss of generality. We then have that $I + XY = \begin{bmatrix} 1 + a - c & c - a \\ c - b & 1 + b - c \end{bmatrix}$. After simplification, its determinant is $det(I + XY) = 1 + a + b - 2c$.

This quantity is strictly positive for $c < \frac{1+a+b}{2}$, which is always the case for as $c^2 \leq ab$. Hence, the matrix is invertible and we have, after computation,

$$M = (I + XY)^{-1}X = \frac{1}{1 + a + b - 2c} \begin{bmatrix} a + ab - c^2 & c + ab - c^2 \\ c + ab - c^2 & b + ab - c^2. \end{bmatrix} \tag{40}$$

Then, $M$ is max-diagonally dominant if and only if $a \geq c$ et $b \geq c$, as expected. Finally, this shows that the embedding $x$ is $Y$-good for any $Y$, hence good. $\qquad\square$

We now focus on the case $A = 2$, for which we obtain a complete characterization of the goodness.

**Proposition 5.** *Consider the GBT model with embedding $x = [x_a, x_b] \in \mathbb{R}^{D \times 2}$ and root law $f$. Then, the model is good if and only if, for any $(a, b) \in \mathcal{A}^2$, $x_a = x_b$ or $x_a^T x_b \leq \min(\|x_a\|^2, \|x_b\|^2)$. The latter is equivalent to*

$$\alpha(x_a, x_b) \leq \alpha_0(\|x_a\|, \|x_b\|) = \arccos\left( \min\left( \frac{\|x_a\|}{\|x_b\|}, \frac{\|x_b\|}{\|x_a\|} \right) \right) \in [0, \pi/2] \tag{41}$$

*where $\alpha(x_a, x_b) = \frac{x_a^T x_b}{\|x_a\| \|x_b\|}$ is the angle between $x_a$ and $x_b$.*

*Proof.* We apply Proposition 4 to $a = x_a^2$, $b = x_b^2$, and $c = x_a x_b$. Then, the goodness is equivalent to $x_a x_b \leq \min(x_a^2, x_b^2)$. This relation is true for $x_a = x_b$ and $x_a x_b < 0$. Otherwise, we have $0 < x_a, x_b$. The relations $x_a x_b \leq x_a^2$ and $x_a x_b \leq x_b^2$ implies that $x_b \leq x_a$ and $x_a \leq x_b$ respectively, leading to a contradiction. Finally, the goodness is equivalent to $x_a = x_b$ or $x_a$ and $x_b$ have different signs.

For the $D$ dimensional case, the same observation holds with $a = \|x_a\|^2$, $b = \|x_b\|^2$, and $c = x_a^T x_b$. Then, Proposition 4 implies that the goodness is equ bivalent to $x_a^T x_b \leq \min(\|x_a\|^2, \|x_b\|^2)$. The angular characterization follows easily. $\qquad\square$

We shall see that the goodness is very restricted for $D = 1$.

**Proposition 6.** *We consider the GBT model with embedding $x = [x_a]_{a \in \mathcal{A}} \in \mathbb{R}^{1 \times A}$, root law $f$, and Laplacian matrix $L = 0$. The model is good if and only if*

$$x = [u, \ldots, u, -v, \ldots, -v, 0, \ldots, 0]^T P = [u 1_{A_1}, -v 1_{A_2}, 0_{A_3}]^T P \tag{42}$$

*for some $u, v > 0$ and $P$ a permutation matrix.*

*Proof.* The goodness is equivalent to the fact that, for any $(ab)$, $(x_a - x_b)x_a \geq 0$ and $(x_b - x_a)x_b \geq 0$. This is equivalent to $x_a x_b \leq min(x_a^2, x_b^2)$, i.e. $x_a = x_b$ or $x_a x_b \leq 0$. This means that any $x_a > 0$ should have a common value $u > 0$ and any $x_b < 0$ should have a common value $-v < 0$. Hence, $x_a$ can take only the values $u, -v$ and $0$. Permuting the values, we obtain (42). $\qquad\square$

## E   Counterexamples for Monotonicity

**Proposition 7.** *There exists good embeddings $x_1$ and $x_2$ such that $x = \begin{bmatrix} x_1 & x_2 \end{bmatrix}^T$ is not good.*

*Proof.* For $A = 3$ and a Gaussian root law $(Y = 3I - J)$, we consider $x_1$ and $x_2$ such that

$$X_1 = \begin{bmatrix} 0 & 0 & 0 \\ 0 & 1 & 1 \\ 0 & 1 & 1 \end{bmatrix} \quad \text{and} \quad X_2 = \begin{bmatrix} 1 & 1 & 0 \\ 1 & 1 & 0 \\ 0 & 0 & 0 \end{bmatrix}$$

Then, $X = x^T x = x_1^T x_1 + x_2^T x_2 = X_1 + X_2 = \begin{bmatrix} 1 & 1 & 0 \\ 1 & 2 & 1 \\ 0 & 1 & 1 \end{bmatrix}$. Then, $x_1$ and $x_2$ are good embedding (e.g. remarking that they are $J$-blocs matrices and using Theorem 2). The matrix $M = (I + XY)^{-1}X$ is given by

$$M = \frac{1}{8} \begin{bmatrix} 3 & 4 & 1 \\ 4 & 8 & 4 \\ 1 & 4 & 3 \end{bmatrix}$$

and verifies $M_{12} > M_{11}$. This contradicts Definition 7 and $x$ is not $Y$-proof for $Y = 3I - J$, therefore not good. $\qquad\square$

**Proposition 8.** *There exists one-hot encodings $x_1$ and $x_2$ such that $x = [x_1 \quad x_2]^T$ is not a good embedding.*

*Proof.* For $A = 5$, let $x_1$ and $x_2$ be one-hot encoding with Gram matrices

$$X_1 = \begin{bmatrix} 1 & 1 & 1 & 0 & 0 \\ 1 & 1 & 1 & 0 & 0 \\ 1 & 1 & 1 & 0 & 0 \\ 0 & 0 & 0 & 1 & 0 \\ 0 & 0 & 0 & 0 & 1 \end{bmatrix} \quad \text{and} \quad X_2 = \begin{bmatrix} 1 & 0 & 0 & 0 & 0 \\ 0 & 1 & 0 & 0 & 0 \\ 0 & 0 & 1 & 1 & 1 \\ 0 & 0 & 1 & 1 & 1 \\ 0 & 0 & 1 & 1 & 1 \end{bmatrix}.$$

Then, $x_1$ and $x_2$ are good embeddings according to Theorem 2. The Gram matrix of the concatenated embedding $x = \begin{bmatrix} x_1 \\ x_2 \end{bmatrix}$ is

$$X = X_1 + X_2 = \begin{bmatrix} 2 & 1 & 1 & 0 & 0 \\ 1 & 2 & 1 & 0 & 0 \\ 1 & 1 & 2 & 1 & 1 \\ 0 & 0 & 1 & 2 & 1 \\ 0 & 0 & 1 & 1 & 2 \end{bmatrix}$$

is not a $Y$-good embedding for the Laplacian matrix

$$Y = \begin{pmatrix} 2 & -1 & 0 & 0 & -1 \\ -1 & 4 & -1 & -1 & -1 \\ 0 & -1 & 1 & 0 & 0 \\ 0 & -1 & 0 & 1 & 0 \\ -1 & -1 & 0 & 0 & 2 \end{pmatrix}.$$

We indeed have that

$$M = (I + XY)^{-1}X \approx \begin{pmatrix} 0.96 & 0.70 & 0.85 & 0.52 & 0.67 \\ 0.70 & 0.90 & 0.95 & 0.67 & 0.68 \\ 0.85 & 0.95 & 1.48 & 0.83 & 0.84 \\ 0.52 & 0.67 & 0.83 & 1.08 & 0.56 \\ 0.67 & 0.68 & 0.84 & 0.56 & 0.93 \end{pmatrix}$$

is such that $M_{23} > M_{22}$. □

## F    Results of Section 4.2

We now show that any embedding can be made $Y$-good by appending a sufficiently dominant identity component. This result guarantees that, asymptotically, adding uncorrelated features improves embedding monotonicity, regardless of the original embedding structure. This can be made in relation with Figure 1 for which we compare i.i.d. Gaussian $x$ with it's concatenation with $I$.

**Proposition 9.** *For any embedding $x$ and any Laplacian matrix $Y$, the embedding $x_\lambda = [I \quad x/\lambda]^T$ is $Y$-good for any $\lambda > 3\sqrt{A}\|x^T x\|/\text{DiagDom}(\sigma^2 Y)$ where $\text{DiagDom}(Y) = \min_{(ab)}(I + Y)_{aa}^{-1} - (I + Y)_{ab}^{-1} > 0$.*

*Proof of Proposition 9.* We set $M(X, Y) = (I + XY)^{-1}X$. The Frobenius norm is such that $\|X + Y\| \le \|X\| + \|Y\|$ and $\|XY\| \le \|X\|\|Y\|$. Assuming that $\|X\| < 1$ and for $Z = (I + X)^{-1} - (I - X)$, we have

$$\|Z\| \le \|X\|^2 \sum_{k \ge 0} \|X\|^k = \frac{\|X\|^2}{1 - \|X\|}. \tag{43}$$

The matrix $I + X/\lambda$ is positive definite, hence invertible and we have

$$M(I + X/\lambda, Y) = (I + (I + X/\lambda)Y)^{-1}(I + X/\lambda) = ((I + X/\lambda)^{-1} + Y)^{-1}. \tag{44}$$

Then, there exist matrices $Z$ and $W$, that we will control later on, such that

$$M(I + X/\lambda, Y) = (I - X/\lambda + Z + Y)^{-1} \tag{45}$$

$$= (I + Y)^{-1}(I + (Z - X/\lambda)(I + Y)^{-1})^{-1} \tag{46}$$

$$= (I + Y)^{-1}(I + (Z - X/\lambda)(I + Y)^{-1} + W). \tag{47}$$

From now, we assume that $\lambda > 3\sqrt{A}\|X\|$. In particular, using that $h : x \mapsto \frac{x}{1-x}$ is increasing over $(0, 1)$ we have that $\frac{\|X\|/\lambda}{1 - \|X\|/\lambda} \leq h(\sqrt{A}/3) \leq h(1/3) = \frac{1}{2}$. According to (43) applied to $X/\lambda$, we therefore have

$$\|Z\| \leq \frac{\|X/\lambda\|^2}{1 - \|X/\lambda\|} \leq \frac{\|X\|}{2\lambda}. \tag{48}$$

We also have that

$$\|X/\lambda - Z\| \leq \frac{\|X\|}{\lambda} + \frac{\|X\|}{2\lambda} = \frac{3\|X\|}{2\lambda}. \tag{49}$$

Note that $\frac{3\|X\|}{2\lambda} \leq \frac{1}{2}$ since $\lambda > 3\sqrt{A}\|X\|$ and we can apply (43) to evaluate

$$\|W\| \leq \frac{\|(X/\lambda - Z)(I + Y)^{-1}\|^2}{1 - \|(X/\lambda - Z)(I + Y)^{-1}\|} \leq \frac{A\|(X/\lambda - Z)\|^2}{1 - \sqrt{A}\|X/\lambda - Z\|} = \sqrt{A}\|X/\lambda - Z\|h(\sqrt{A}\|X/\lambda - Z\|) \tag{50}$$

$$\leq \sqrt{A}\frac{3\|X\|}{2\lambda}h\left(\frac{3\sqrt{A}\|X\|}{2\lambda}\right) \leq \frac{3\sqrt{A}\|X\|}{2\lambda}h(1/2) = \frac{3\sqrt{A}\|X\|}{2\lambda} \tag{51}$$

where we used $\|(X/\lambda - Z)(I + Y)^{-1}\| \leq \|X/\lambda - Z\|\|(I + Y)^{-1}\| \leq \sqrt{A}\|X/\lambda - Z\|$ (since $(I + Y)^{-1}$ has eigenvalues smaller than 1) and $h(1/2) = 1$.

Starting again with (45), we then have

$$\|M(I + X/\lambda, Y) - (I + Y)^{-1}\|_\infty \leq \|M(I + X/\lambda, Y) - (I + Y)^{-1}\| \tag{52}$$

$$= \|(Z - X/\lambda)(I + Y)^{-1} + W\| \leq \frac{3\sqrt{A}\|X\|}{2\lambda} + \frac{3\sqrt{A}\|X\|}{2\lambda} \tag{53}$$

$$= \frac{3\sqrt{A}\|X\|}{\lambda}. \tag{54}$$

Let $\mathrm{DiagDom}(Y) = \min_{(ab)}\left((I + Y)_{aa}^{-1} - (I + Y)_{ab}^{-1}\right)$ which is strictly positive since $(I + Y)^{-1}$ is strictly max-diagonally dominant [10]. We therefore have that, for any $\lambda > 3\sqrt{A}\|X\|/\mathrm{DiagDom}(Y)$,

$$\|M(I + X/\lambda, Y) - (I + Y)^{-1}\|_\infty \leq \mathrm{DiagDom}(Y)$$

and therefore $M(I + X/\lambda, Y)$ is max-diagonally dominant, as expected. $\qquad\square$

## G  Inverse of super-Laplacian

**Proposition 10.** *Let $\Delta$ be a super-Laplacian matrix, then for any nodes $a \neq b$ in $\mathcal{A}$, we have*

$$e_a^T \Delta^{-1} e_{a \ominus b} = (\Delta^{-1})_{aa} - (\Delta^{-1})_{ab} \geq 0.$$

*Proof.* We prove the result by interpreting the coefficients of $\Delta^{-1}$ as a probability of some sample path of a discrete-time Markov process on the alternatives. Let $D$ be the diagonal of $\Delta$, and $P$ the matrix defined by

$$\Delta = D(I - P). \tag{55}$$

The matrix $P$ is a row-sub-stochastic matrix. Explicitly,

$$P_{aa} = 0, \qquad\qquad P_{ab} = \frac{|\Delta_{ab}|}{\Delta_{aa}}, \qquad\qquad \sum_{b \in \mathcal{A}} P_{ab} < 1. \tag{56}$$

Let $\bullet$ be an extra symbol and $\mathcal{A}_\bullet = \mathcal{A} \sqcup \{\bullet\}$. Let $\kappa_a = \Delta_{aa} - \sum_{b \neq a} |\Delta_{ab}| > 0$. We define a Markovian random walk on $\mathcal{A}_\bullet$ by the transition matrix $T$:

$$T(b|a) = P_{ab} = \frac{|\Delta_{ab}|}{\Delta_{aa}}, \quad T(\bullet|a) = 1 - \sum_{b \in \mathcal{A}} P_{ab} = \frac{\kappa_a}{\Delta_{aa}}, \quad T(a|\bullet) = 0, \quad T(\bullet|\bullet) = 1. \quad (57)$$

Intuitively, this Markovian process walks over the alternatives according to the weights $|\Delta_{ab}|$, and at each step has a non-zero probability to end in the cemetery $\bullet$.

Now,

$$\Delta_{ba}^{-1} = \left((1-P)^{-1} D^{-1}\right)_{ba}, \quad \Delta_{ba}^{-1} = \sum_{n \geq 0} \left(P^n D^{-1}\right)_{ba}, \quad \Delta_{ba}^{-1} = \sum P_{ba_1} \dots P_{a_{n-1}a} \frac{1}{\Delta_{aa}}. \quad (58)$$

Therefore,

$$\Delta_{ba}^{-1} \kappa_a = \sum T(a_1|b) \dots T(a|a_{n-1}) T(\bullet|a). \quad (59)$$

In other words, $\Delta_{ba}^{-1} \kappa_a$ is the probability that $a$ is the last alternative to be visited before the random walk is killed, given that it started at $b$. Notice that the alternative $a$ may be visited multiple times in those paths. Actually, any path that starts at $b$ and visits $a$ before being killed can be decomposed as the gluing of a path that starts at $b$ and reaches $a$, followed by a path that starts at $a$ and eventually revisits $a$ as its last step before being killed. Therefore,

$$\Delta_{ba}^{-1} \kappa_a \leq \Delta_{aa}^{-1} \kappa_a. \quad (60)$$

Since $\kappa_a > 0$, and since $\Delta$ is symmetric, we finally obtain

$$\Delta_{aa}^{-1} \geq \Delta_{ab}^{-1}. \quad (61)$$

$\square$

## H   Proof of Theorem 2

*Proof.* Fix an arbitrary $\lambda > 0$, and let $\mu = s^2 + \lambda$. There exists a permutation matrix $P$, and an integer partition $A_1 + \dots + A_k = A$, such that

$$X \triangleq \begin{pmatrix} x^T & sI \end{pmatrix} \begin{pmatrix} x \\ sI \end{pmatrix} + \lambda I \quad (62)$$

$$= x^T x + \mu I \quad (63)$$

$$= P \cdot \left(\mu I + \text{block\_diagonal}(J_{A_1}, \dots, J_{A_k})\right) \cdot P^{-1} \quad (64)$$

$$= P \cdot \text{block\_diagonal}(\mu I_{A_1} + J_{A_1}, \dots, \mu I_{A_k} + J_{A_k}) \cdot P^{-1} \quad (65)$$

where every $J_{A_i}$ is a matrix of size $A_i \times A_i$ with all its entries set to 1. Up to renaming the alternatives, we can assume, without loss of generality, that $P = I$.

We notice that the all-one matrix $J$, say of size $A$, satisfies $J^2 = AJ$, and

$$(\mu I + J) \frac{1}{\mu} \left(I - \frac{1}{A + \mu} J\right) = 1 \quad (66)$$

Therefore, $X$ is invertible and

$$X^{-1} = \text{block\_diagonal}\left(\frac{1}{\mu}\left(I_{A_1} - \frac{1}{A_1 + \mu} J_{A_1}\right), \dots, \frac{1}{\mu}\left(I_{A_k} - \frac{1}{A_k + \mu} J_{A_k}\right)\right) \quad (67)$$

Thus, $X^{-1}$ is super-Laplacian. This proves that $\begin{pmatrix} x & sI \end{pmatrix}^T$ is a diffusion embedding. The monotonicity of $\text{GBT}_{f,\sigma,x,L}$ follows from Theorem 1. $\square$

