# OpenReview forum: "Generalizing while preserving monotonicity in comparison-based preference learning models"
_NeurIPS.cc/2025/Conference — NeurIPS 2025 poster_

### Official Review · Reviewer_DMrc · 2025-06-29

**Clarity:** 3
**Significance:** 3
**Originality:** 3
**Rating:** 5
**Confidence:** 3

**Summary:**

The paper addresses to key problems in preference-learning models, monotonicity and score-generalization.
In particular, the authors tackle both by extending a well-established model (GBT) to include embeddings of the alternatives and a more general regularization term.
The authors then prove the conditions necessary to make their extended model monotonic and show how every diffusion embedding as defined in def5 satisfies these conditions.
Finally, the authors perform a limited experiment to showcase their model against GBT.










The paper addresses the problem of monotonicity, or lack thereof, in preference learning models.
In particular, the authors propose a new class of preference learning algorithms, based on the Generalized Bradly-Terry model, which also considers embeddings of the alternatives.
With appropriate conditions on the abovementioned embeddings the authors show that they can guarantee monotonicity.

**Questions:**

Major questions:
See the weaknesses above.

Clarifications:
- L117--120: There might be a contradiction here. If EXCHANGE replaces $(a_n, b_n, r_n)$ with its opposite $(a_n, b_n, -r_n)$, I don't see why preference algorithms should interpret the two comparison samples identically. At the limit, one can exchange all of the comparison samples with their opposites, most certainly affecting training.
- L140: The authors should provide proof that $\preceq_a$ is indeed a preorder; in particular, I am skeptical about reflexivity, as I cannot see a way to define an operation (infinitely) favors $a$ while leaving a dataset $\mathbf D$ unchanged.
- L163: In what sense is proposition 1 a generalization of [9]?
- L274: How does $H$ depend on $L$? The sentence build like the two are indeed related, but I cannot see how.
- L284: Shouldn't $x$ be $(L + \tilde H)$-good?
- L467: Based on eq7, shouldn't it be $\mathcal L_\lambda(\beta|\mathbf D, o) - \mathcal L(\beta|\mathbf D) = \lambda \cdot (\Phi_f(\theta_{ab}) - r\theta_{ab})$?

Spelling and notation:
- L56: "monotony" should be "monotonicity; this may have happened in other parts of the manuscript.
- L107, eq3: $\mathcal R$ is used with different meanings.
- L116: consider reserving $\mathcal O$ for its usual meaning in asymptotic analysis.
- L130, eq6, eq7, L262: the notation for UPDATE is inconsistent; this might be the case for the other operations as well.
- L139 (and others): Given the definition of $\Omega$, it's equivalent (and formally correct) to replace $\geq$ with $\in$.
- L149: ",." at the end.
- eq3: $\theta_b(\beta)$ is not defined; is it $\theta(\beta)_b$? The first symbol is not defined.
- eq4: $x_{ab}^T\beta$ is confusing for two reasons: (1) if it means $(x^T)\_{ab}$  why not use $x_{ba}$? (2) $\beta$ was defined on L181 as a vector and $x^T_{ab}$ looks like a scalar, meaning that $x_{ab}^T\beta \in \mathbb  R^D$, which is in conflict with the rest of the terms in the equation.
- L255, 226: proposes, provides.
- L250: $o_\lambda$ is not defined.
- eq9: What is $e$? L305 defines $I$ as the identity matrix.
- The figures are not saved in a vector format, making the labels hard to read when the paper is printed out.

**Ethical Concerns:**

["NO or VERY MINOR ethics concerns only"]

**Final Justification:**

The criticalities I found in the paper were mainly about clarity (notation) and the weak experimental comparison provided in Section 5. After discussing with the authors, I am now satisfied by the improvements made to the notation and am now convinced that the experimental setup is appropriate for publication, although far from perfect.

I believe the theory developed in the paper is a worthy addition to the field and therefore recommend to accept the paper.

**Limitations:**

The authors discuss some limitations of their model, especially regarding the extension to non-diffusion embeddings.
What is missing here, in my opinion, is a discussion on the drawbacks stemming from trying to use the model in practice (W3).

**Paper Formatting Concerns:**

None.

**Quality:**

3

**Strengths And Weaknesses:**

## S1 - Interesting model
Extending GBT to allow for score-generalization is potentially impactful for the community, as it addresses a strong motivation.
Also, the fact that every diffusion embedding can be used within the model potentially makes it flexible in tackling different scenarios.

## W1 - Notation
The mathematical notation is often inconsistent and confusing, making it quite hard to follow the text and, especially, the proofs.
Notable issues include:
1. The indiscriminate usage of the same letter $\theta$ for both a matrix and a vector; both $\theta_{ab}$ and $\theta_a$ indicate a real number in different parts of the text.
2. An inconsistent usage of typefacing. Matrices are sometimes indicated with lowercase letters ($x$), greek letters ($\theta$), and uppercase letters ($L, H$). On the contrary, the same typeface is used for multiple different kinds of objects; examples are $A, D, N \in \mathbb N$ and $L$ being a matrix, as well as $\beta$ being a matrix and $\sigma$ being a real number.
3. Undefined symbols, such as $e_{ab}$ in Lemma 1; I am guessing $e$ is the identity matrix of some dimension, but in Definition 3 the symbol $I$ was introduced for that.
4. Inconsistencies in Lemma 1 and its proof, where both $\frac{d\theta^\*(\lambda)}{d\lambda}$ (eq9) and $\frac{d\theta^*\_\lambda}{d\lambda}(\mathbf D, o)$ (eq30) are used to (potentially?) indicate the same object.

## W2 - Experiments
The experimental evaluation is insufficient to showcase the benefits and potential drawbacks of the proposed model over other approaches.
In my opinion, both of the following issues with the experimental setup stem from W3.

### W2.1 Competitors
The experiments omit comparisons with other potential competitors such as the ones described in L79--87.

### W2.2 Data-generating process
To my understanding, the data-generating process described in Section 5.2 aligns perfectly with the proposed model's assumptions, making it a very unfair comparison ground for standard GBT.

## W2.3 - Error bars
The authors report in the checklist that they are purposely not reporting error bars in their figures, as they are considering an arguably large number of repetitions (100). I believe error bars (preferably, standard deviation) should instead be included, as it helps discerning how significant the differences between the models' performances are.
Ideally, some statistical analysis of the results would also help.

## W3 - Assumptions of the model
As described in Section 3, the model assumes access to a known embedding matrix $x$, which is a strong and unrealistic assumption in most practical scenarios.
The sensitivity of the method to this assumption is evident in Figure 2a, where not choosing the “correct” embedding lead to significant performance degradation. The "correct" embedding is the one aligning with the data-generating process.
Without a mechanism to learn or robustly adapt this embedding, the practical utility of the method remains limited.
I also wonder, if learning the embedding is possible, whether that would make the optimization problem intractable or very hard to optimizer, given the increased number of variables in the model.

---

> ### Author Rebuttal · Authors · 2025-07-31
>
> We thank the reviewer for the time they took to review our paper, and for their detailed remarks.
>
> ## W1 - Notation
>
> > [...] Issues include:
>
> > The indiscriminate usage of the same letter \theta for both a matrix and a vector; both \theta_{ab} and \theta_a indicate a real number in different parts of the text.
>
> The variable $\theta$ is introduced in section 2.3, (eq. 2) states that $\theta \in \mathbb{R}^A$. We admit that the notation $\theta_{ab} = \theta_a - \theta_b$, specific to BT and GBT literature, is confusing. We will update the paper accordingly.
>
> Variable $\theta$ should not be a matrix. We have checked against this, but did not find occurences. Could you please point to places where you understood that $\theta$ was a matrix?
>
> > An inconsistent usage of typefacing. [...]
>
> Indeed, we have not strictly followed a typesetting convention, and we understand that this may reduce the paper readability for people used to such a convention. We can add a "Notation" table or section that summarizes our notations.
>
> > Undefined symbols, such as e_{ab} in Lemma 1 [...]. Inconsistencies in Lemma 1 and its proof [...].
>
> Thanks for pointing this out. We added in lemma 1 the following "There, $e_a$ denotes the $a$-th vector of the cartesian basis, and $e_{ab} = e_a-e_b$", and have fixed notation (eq9) and (eq30).
>
> ## W2 - Experiments
>
> ### W2.1 Competitors
>
> > The experiments omit comparisons with other potential competitors such as the ones described in L79--87.
>
> Let us stress that the main interest, and challenge, of this work is to guarantee the monotonicity of the learned model. Alternative methods of l.79-87 (Deep neural network generalizations of BT, RLHF, DPO, etc) do not benefit from any monotonicity guarantee. Worse, forms of non-monotonicity was observed during the finetuning of LLMs; see eg [Hong et al.; Orpo [...], 2024].
> Hence our choice to limit our numerical illustrations to methods with monotonicity guarantees.
>
> ### W2.2 Data-generating process
>
> > To my understanding, the data-generating process described in Section 5.2 aligns perfectly with the proposed model's assumptions, making it a very unfair comparison ground for standard GBT.
>
> The matrix $x$ used by the learning model indeed corresponds to the matrix of the data-generating process.
> GBT with embedings thus performs drastically better than GBT. While that may be seen as unfair, this also speaks to the interest of this contribution: adding contextual, or descriptive, information to the GBT model, that until now could only leverage comparisons.
> In view of the well- vs. mis-specified results in statistics, we believe that replacing the exact embedding matrix by an approximation would degrade the performance the GBT with embedding model in a limited way, thus maintaining it above GBT. We detail this point in W3.
>
> ### W2.3 - Error bars
>
> > The authors report in the checklist that they are purposely not reporting error bars in their figures, as they are considering an arguably large number of repetitions (100). I believe error bars (preferably, standard deviation) should instead be included [...].
>
> We understand that plots with error bars showing standard deviations are standard in the ML community. We have produced such plots.
> Nevertheless, we cannot report them here nor provide a link to our images, given the rules for this year's rebuttal format. We report a table version of these plots below.
>
> Figure 2a. nMSE as a function of D for A=25 alternatives and N=500 comparisons over 100 seeds.
> The following table reports mean ± 1.96 * stdev / sqrt(n_seeds).
> | Dimension D |  GBT & encoding  |       GBT        |     Encoding     |
> |-------------|------------------|------------------|------------------|
> |      2      | 0.0497 ± 0.00605 | 0.0801 ± 0.00676 |  0.39 ± 0.0432   |
> |      4      | 0.0369 ± 0.00449 | 0.0855 ± 0.00614 |  0.248 ± 0.0314  |
> |      8      | 0.0269 ± 0.00249 |  0.102 ± 0.0079  |  0.125 ± 0.0124  |
> |     16      | 0.0182 ± 0.00163 |  0.191 ± 0.0118  | 0.0377 ± 0.00357 |
> |     20      | 0.0183 ± 0.00169 |   0.2 ± 0.0112   | 0.0271 ± 0.00274 |
>
> Figure 2b. nMSE with respect to the number of comparisons N for A=20, D=10, and 1000 seeds.
> The following table reports mean ± 1.96 * stdev / sqrt(n_seeds).
> |  N  |  GBT & encoding   |       GBT       |
> |-----|-------------------|-----------------|
> |  1  |  0.934 ± 0.00974  | 0.983 ± 0.00125 |
> | 71  |  0.113 ± 0.00348  | 0.454 ± 0.00444 |
> | 141 |  0.0636 ± 0.0019  | 0.29 ± 0.00426  |
> | 191 | 0.0508 ± 0.00145  | 0.231 ± 0.0039  |
> | 291 | 0.0301 ± 0.000989 | 0.182 ± 0.00378 |
>
> Note that the values reported above differ slightly from the plots. The reason is that we have improved our codebase (fixing MLE optimization and unstable gradient). Importantly, the behavior of the curves remain identical.
>
> We believe that the main contributions of the paper is extending GBT with (1) embeddings, so that it may generalize, and (2) Laplacian regularization, to account for known relationships all the while maintaining the crucial property of monotonicity.
> We agree that a statistical analysis is desirable in view of the experiments, but feel that it would not directly support our message.
>
> ## W3 - Assumptions of the model
>
> > As described in Section 3, the model assumes access to a known embedding matrix x, which is a strong and unrealistic assumption in most practical scenarios. [...]
>
> In this work, we do not study the statistical performance of the proposed extension of GBT; we rather focus on monotonicity, which is the content of Section 3.These monotonicity guarantess holds independently of how the data is generated.
>
> We agree that statistical performance is interesting. We peek at the statistical performance of our extension of GBT against GBT in the experiments, but leave thourough experiments for the statistical study of these estimators. Again, we believe it is outside the scope of our paper, which focuses on monotonicity, a critical property for preference learning, and one that receives little attention.
>
> Note that we are currently investigating statistical aspects. Preliminary results indicate that the statistical performance of the GBT with embeddings model degrades in a controlled manner when the embedding matrix $x$ is increasingly distant from the true embedding $x^\dagger$. Learning the embedding matrix seems possible with, eg, hierarchical bayesian learning.
>
> ## Clarifications
>
> > L117--120: There might be a contradiction here [relative to EXCHANGE].
>
> Thank you, this is a typo. We meant: EXCHANGE replaces $(a_n, b_n, r_n)$ with $(b_n, a_n, -r_n)$.
> We thus consider in this work preference learning algorithms that treat (a,b,r) and (b,a,-r) in the same way. Note that this assumption is valid for BT, GBT, and the proposed generalization of GBT.
>
> > L140: The authors should provide proof that $\preceq_a$ is indeed a preorder [...].
>
> We have added the following setence after the claim that $\succeq$ is a preorder:
> "Indeed, $\preceq_{a}$ is reflexive: any dataset $\mathbf{D}$ equals $o(\mathbf{D})$ with $o = UPDATE_{n, r}$ with $n=1$, and $r=r_1$. The relation $\preceq_{a}$ is transitive: if $\mathbf{D}_1 \preceq_a \mathbf{D}_2$ and $\mathbf{D}_2 \preceq_a \mathbf{D}_3$, then there exists operations $o_1$ and $o_2$ that favor $a$, such that $bD_1 = o_1(\mathbf{D}_2)$, and $\mathbf{D}_2=o_2(\mathbf{D}_3)$; thus $\mathbf{D}_1=o_1 \circ o_2(\mathbf{D}_3)$, where $o_1\circ o_2$ is an operation that favors $a$ by definition 1 so that $\mathbf{D}_1 \preceq_a \mathbf{D}_3$."
>
> > L163: In what sense is proposition 1 a generalization of [9]?
>
> Theorem 2 of [9] guarantees monotonicity of $GBT_{f, \sigma}$ for datasets which contain at most one comparison between any two elements $a$ and $b$. In Theorem 3, we provide monotonicity for datasets which may contain repeated repetitions between $a$ and $b$. We have reprased Proposition 1 to make this clear.
>
> > L274: How does $H$ depend on $L$? [...]
>
> Indeed, $H$ does not depend on $L$. The matrix $L$ is used below (L279), and should be declared there.
> We have reformulated as follows:
> - L274: "Note first that, the Hessian $H$ of [...] is also a Laplacian matrix".
> - L279: "Therefore, given any prior Laplacian matrix $L$, the matrix $L + \tilde{H}$, [...]".
>
> > L284: Shouldn't $x$ be $(L + \tilde H)$-good?
>
> This is a typo, thank you. We have reformulated, at line 284: "This is precisely implied by $x$ being good."
>
> > L467: [missing $\lambda$ at (eq7)]
>
> A factor $\lambda$ was indeed missing, we have added it. Thank you.
>
> ## Spelling and notation
>
> > [..] notation for UPDATE is inconsistent [...]
>
> We have fixed the notation of UPDATE and APPEND, they are now consistent.
>
> > L139 (and others): Given the definition of $\Omega$, it's equivalent (and formally correct) to replace $\geq$ with $\in$.
>
> We do not understand. There is no $\Omega$ or $\geq$ at l.139. Can you provide precisions please?
>
> > eq3: $\theta_b(\beta)$ is not defined; is it $\theta(\beta)_b$? The first symbol is not defined.
>
> Indeed. We have added a sentence that clarifies the definition of $\theta$ as a function of $\beta$ into $\mathbb{R}^A$, and that explains $\theta_b(\beta)$.
>
> > eq4: $x_{ab}^T\beta$ is confusing [...]
>
> We admit the notation is confusing. We have clarified in the text that $x_a \in \mathbb{R}^D$ is the $a$-th column of $x$, for any $a \in [A]$ at line 182, and that $x_{ab} = x_a - x_b$.
>
> > eq9: What is $e$? [...]
>
> In equation 9, $e_a$ denotes the $a$-th vector of the cartesian basis of $\mathbb{R}^A$, and $e_{ab} = e_a - e_b$. We have added this explanation in the lemma.
>
> > The figures are not saved in a vector format [...]
>
> We updated figures to vector format, and made the labels more readable.
>
> We have fixed the remaining comments, thank you.
>
> We hope to have addressed all your concerns. We remain at your disposal may you have any further questions or require additional information. We would be grateful if you could consider revising your score based on the answers we provided.

---

> ### Comment · Reviewer_DMrc · 2025-08-02
>
> I thank the authors for the extensive and comprehensive rebuttal, it really helped understanding some parts of the contribution.
> I still feel the need to further discuss some of the points.
>
> ### W1
> > Variable $\theta$ should not be a matrix. We have checked against this, but did not find occurences. Could you please point to places where you understood that $\theta$ was a matrix?
>
> Intuitively and informally, a matrix is "an object with two indices"; as opposed to vectors, which only have one index. As $\theta$  appears both with two indices and with one index (see, for instance, eq2), I interpret it sometimes as a vector and sometimes as a matrix.
> Although one can infer from the context what interpretation of $\theta$ is correct, having a different symbol (such as $\Theta_{ab}$ instead of $\theta_{ab}$ ) could fix this problem.
>
> I don't want to emphasize this point: as the authors propose, having a clear and exhaustive notation table would already greatly help clarity (although some inconsistencies remain, see my last point in this comment).
>
> ### W2
> I agree with the authors that the main contribution of the paper is to provide an improved monotonic model.
> Now, the monotonicity of the proposed model is guaranteed theoretically.
> What I am missing is to show to what extent the proposed model improves on GBT: it is clearly more general (L193) and better than GBT in a strictly controlled setting (Section 5.2), but is it better in general---for instance, to estimate Youtube scores, where the data generating process is unknown?
>
> In other terms: is the performance improvement worth it to deal with the added complexity of the proposed model?
>
> ### Notation
> I sincerely appreciate the authors' efforts in improving the clarity and preciseness of the notation, and I am truly sorry for my point about L139 (which is non-substantial).
> As the mathematical proof of monotonicity is the central contribution of the paper, a clear and consistent notation is, in my opinion, absolutely crucial for readability and proof-reading.
>
> There is one last point I would like to raise, which is the inconsistency between the usage of indices.
> As the authors acknowledge in the rebuttal, $x$ is a matrix, $x_a$ is the $a$-th column of $x$, and $x_{ab}$ is another column vector---similar considerations apply to the identity matrix $e$, $e_a$, and $e_{ab}$.
> However, the convention is different for $\theta$, where $\theta_a$ and $\theta_{ab}$ are both scalars.
>
> This is what confused me while reading the proof (and eq4 in particular), as I assumed, based on established notation practices and $\theta$, that $x_{ab}$ was a scalar.
> In the light of this, I urge the authors to harmonize their notation and usage of indices (in addition to adding a notation table to the manuscript).

---

> > ### Author Response · Authors · 2025-08-04
> > **Answer to Reviewer DMrc by Authors**
> >
> > We thank the reviewer for providing high quality comments on our submission; we believe this improves the work a great deal.
> >
> > ## W1 and Notation
> >
> > We apologize for the difficulties raised by notations. We acknowledge that there is an overlap between the GBT-specific notation $\square_{ab}$, that denotes $\square_a - \square_b$, and the classical notation $\triangle_{ab}$, that denotes the entry at row $a$ column $b$ of matrix $\triangle$.
> > We followed the first convention out of consistency with the initial GBT paper [9].
> > Thanks to your input, we understand and agree that this notation is confusing.
> >
> > We have thus updated our manuscript:
> > 1. $\square_a - \square_b$ is now always denoted $\square_{a \ominus b}$, and
> > 2. $\triangle_{ab}$ is reserved for the usual matrix indexing.
> >
> > We will complete this by a table that summarizes notations, placed in the introduction.
> >
> > For reference, in the submitted pdf $\theta$ always denotes a vector in $\mathbb{R}^A$, and $\theta_{ab}$ always denotes quantity $\theta_a - \theta_b$; this is indicated at (eq. 2), l. 181, and l. 153; it is not stated elsewhere in the paper.
> > More generally, the meaning of indices $\cdot_{ab}$ in the submitted pdf is
> > - $\square_{ab}$, for $\square$ equal to $\theta$ (a vector), $x$ (a matrix), and $e$ (a matrix). All occurences should all be understood as $\square_a - \square_b$.
> > - $\triangle_{ab}$, for $\triangle$ equal to $L$, $\Delta$, $N$, $G$, $H$, $\Phi''_f$, and $e$. All occurences should all be understood as the element at row $a$ and column $b$.
> >
> > (We will also replace the notation $S_{ab}$ by $S^{ab}$, and $S_{ab}[d, e]$ by $S^{ab}_{de}$ at l. 245 and following.)
> >
> > ## W2
> >
> > > I agree with the authors that the main contribution of the paper is to provide an improved monotonic model. Now, the monotonicity of the proposed model is guaranteed theoretically. What I am missing is to show to what extent the proposed model improves on GBT: it is clearly more general (L193) and better than GBT in a strictly controlled setting (Section 5.2), but is it better in general---for instance, to estimate Youtube scores, where the data generating process is unknown?
> >
> > > In other terms: is the performance improvement worth it to deal with the added complexity of the proposed model?
> >
> > We believe that *the added complexity is rather mild*, especially relative to Deep Learning based methods.
> > With no Laplacian regularization, the only difference with the GBT estimator is the aditional embedding matrix $x$ in the optimization problem (eq 2). The optimization problem remains very well behaved, namely strongly convex, and easy to optimize even when it is high-scale to huge-scale. For instance, one can use state-of-the-art off-the-shelf methods such as gradient descent or coordinate-descent methods equipped with inertia, or quasi-Newton methods.
> >
> >
> > Besides, *the potential for improvement is great*, especially in the setting of estimating Youtube scores [13, 14].
> > Indeed, in that situation, there are two challenges: (1) identifying the best videos among the videos seen by a user, so that they may be recommended to others, and (2) recommend new videos to a user that they are likely to like.
> > The GBT model assigns zero score to uncompared videos. Thus, there is no basis to suggest a new video to that user.
> > To put it another way, if we conduct an experiment to fit GBT on a training set (compared videos),
> > we will naturally see GBT perform very poorly on the test set (uncompared videos).
> > Adding embeddings, that is descriptive information on videos provides a way to score yet-uncompared videos.
> > Typically, consider the one-hot embedding that assigns to each video the identifier of its content creator.
> > Assuming a user has favorably compared some videos of a given content creator, that embedding would allow suggesting high scores to yet-to-be-watched videos of that same content creator.
> >
> > We believe that the ability of GBT with Embeddings (GBTD) to predict scores for unseen alternatives while preserving monotonicity is what makes it worth the added mild complexity relative to GBT.
> >
> > We are currently looking into statisctical aspects in theory, and into how GBT and GBTD compare in practice, with data of [13, 14].
> > At the current early stage, it seems that the GBTD estimator is (1) useful in practical scenarii most of the time, and that (2) in theory, GBTD is not uniformly better than GBT: for instance, two items may have similar embeddings but highly different preferences by the user. It would then take more comparisons for GBTD than for GBT to reach similar estimations of scores, that is GBTD may have a worse sample complexity than GBT in specific, adversarial, situations.
> >
> > We hope to have addressed the concerns W1, W2, and Notation. We remain at your disposal may you have any further questions or require additional information. We would be grateful if you could consider revising your score based on the answers we provided.

---

> > > ### Comment · Reviewer_DMrc · 2025-08-06
> > >
> > > I would recommend, even if partial, to include the experimental comparison between GBT and GBTD in the supplementary material (or, at least, to GitHub), as it can help future readers assess the practical utility of GBTD.
> > >
> > > Apart from this, thank you for your detailed and constructive answers, all of my other concerns have been properly addressed and I have revised my score accordingly.

---

### Official Review · Reviewer_XoXh · 2025-07-02

**Clarity:** 3
**Significance:** 3
**Originality:** 3
**Rating:** 5
**Confidence:** 4

**Summary:**

The authors consider the problem of generalizable preference learning from pairwise comparisons. They focus on a desirable monotonicity property, which holds for a preference learning algorithm if observing data that favors an item $a$ only causes the learned value of $a$ to increase. Surprisingly, Bradley-Terry with linear utilities violates monotonicity. However, the authors show that if item embeddings satisfy certain properties, then a generalized Bradley-Terry model (GBT) does satisfy monotonicity. In simulation experiments, the authors examine the probability that random Gaussian embeddings satisfy their "goodness" property (unfortunately, this is small when there are few embedding dimensions relative to items). In experiments on synthetic data, they demonstrate how generalization across items is achieved through the model's Laplacian regularization and item embeddings.

**Questions:**

1. If I'm understanding the definition of monotonicity correctly, any dataset operation that favors $a$ should increase the learned score score $\theta_a$ of $a$. Is this sufficient to cover the "buggy" example in the introduction? Suppose adding a comparison where $a$ beats $b$ does increase the score of $a$, but it also increases the score of $c$ by even more, causing $a$ to now perform worse against $c$. (I know this is a weird learning algorithm, but perhaps $c$'s embedding indicates it has all the good features of $a$ and none of the bad features of $b$.) How is this avoided by monotonicity?

2. Is there some intuition for what embeddings are diffusion embeddings? Are the embeddings in the introduction example not diffusion embeddings, and is this possible to tell without computing inverses?

3. What parts of the model are required for monotonicity? Can we get rid of the $L_2$ regularization? The Laplacian regularization?

Minor:
- line 56: "monotony"
- Line 68: another relevant choice citation is McFadden's multinomial/conditional logit
- Line 149: has an extra comma
- Line 162: this should be referencing Proposition 1, right?
- $\mathcal R$ is used both for the set of comparison scores and the regularization term
- Smoothed rather than smoothened?

**Ethical Concerns:**

["NO or VERY MINOR ethics concerns only"]

**Final Justification:**

I only had some minor questions/comments/concerns and continue to feel that this is a nice paper, well-reflected by a 5.

**Limitations:**

Nice concise limitations section. I don't see this work having negative societal impacts.

**Paper Formatting Concerns:**

Numerical references are currently used as nouns; I think these should be "Author [1]" citations rather than just "[1]" (e.g., lines 68 and 72).

**Quality:**

3

**Strengths And Weaknesses:**

Strengths:
1. The contribution is clear and significant
2. The writing and notation are good
3. The problem is timely and interesting

Weaknesses:
1. A few minor points that could be made clearer (see questions)

Quality: Good quality, well situated in the literature, good mix of theory and experiments, well-structured.

Clarity: The clarity is high overall. A few points could be better explained (see questions)

Significance: Important and relevant to preference learning.

Originality: The model itself uses standard techniques (utility = linear function of features, Laplacian regularization), but the properties of embeddings required for monotonicity and the associated analysis are original to my knowledge. I'm not sure I'd call GBT with diffusion priors a "new model"; e.g., see (1) https://doi.org/10.1017/nws.2023.20 and (2) https://arxiv.org/abs/1703.07520 for other work applying item features and Laplacian regularization to learning preference models. The only difference between GBT with diffusion priors and the Laplacian-regularized conditional logit model in (1) is the use of the comparison score $r$ rather than a fixed choice of preferred item (and the focus here on pairwise comparisons). I don't think this is an issue; the major contribution of this work is the analysis of monotonicity. I would just tone down the claimed novelty of the model.

---

> ### Author Rebuttal · Authors · 2025-07-31
>
> We thank the reviewer for the time they took to review our paper, and for their remarks.
>
> > If I'm understanding the definition of monotonicity correctly, any dataset operation that favors $a$ should increase the learned score $\theta_a$ of $a$. Is this sufficient to cover the "buggy" example in the introduction? Suppose adding a comparison where $a$ beats $b$ does increase the score of $a$, but it also increases the score of $c$ by even more, causing $a$ to now perform worse against $c$. (I know this is a weird learning algorithm, but perhaps $c$'s embedding indicates it has all the good features of $a$ and none of the bad features of $b$.) How is this avoided by monotonicity?
>
> This is a very interesting remark. It seems that you formulate an even stronger monotonicity property: if $a$ is favored over $b$ then the score of $a$ should increase *and* no other alternatives increases by a larger amount. We have not considered this formulation. However, because our model relies on diffusion, we believe this property is likely to be true for GBT with diffusion embeddings and laplacian regularization. The intuition is that favoring $a$ over $b$ is like heating $a$ and cooling $b$ (both by a unit amount) and letting the heat diffuse throughout the (similarity) network, so no other alternative should have its "temperature" (score) vary by a larger amount. Of course, this statement is not a proof, and we would need to check it. Thank you for your input.
>
> > Is there some intuition for what embeddings are diffusion embeddings? Are the embeddings in the introduction example not diffusion embeddings, and is this possible to tell without computing inverses?
>
> This is actually a question we decided to study after the submission of the present paper. More precisely, as you noticed, diffusion embeddings (and good embeddings) are defined in an algebraic way. We have recent unpublished results that aim at characterizing diffusion embeddings in a combinatorial way. This topic is complex enough to be difficult to discuss in more details here, and to make a contribution on its own. So, if that is okay with you, we would like to stay at that description.
>
> > What parts of the model are required for monotonicity? Can we get rid of the $L_2$ regularization? The Laplacian regularization?
>
> Monotonicity requires that embeddings be "good", as per Definition 7, and $\sigma > 0$.
> The Laplacian regularization defined by matrix $L$ can be removed without hindering monotonicity, as the null matrix is a Laplacian matrix.
> The $\ell_2$ regularization cannot be discarded. Indeed, it is crucial for the GBT estimator to be well-defined (in particular, unique), even without embeddings in the initial GBT work [9]. In addition, the regularization parameter $\sigma$ plays a key role in the technical Lemma 1.
>
> > Minor:
>
> > Line 162: this should be referencing Proposition 1, right?
>
> Our intention was for Proposition 1 to recall [Th. 2, 9], and mention that the forthcoming Th. 3 generalizes this result. A subtlety is that the original GBT [9] assumes that two elements (a, b) are compared at most one time, while we account for more that one comparison between (a, b) in the dataset.
> We have updated the sentence to clarify this subtlety.
>
> > [other comments]
>
> Thank you for taking the time to note these points. We have addressed them right away in the manuscript.
>
> We hope to have addressed all your concerns. We remain at your disposal may you have any further questions or require additional information. We would be grateful if you could consider revising your score based on the answers we provided.

---

> > ### Comment · Reviewer_XoXh · 2025-08-01
> >
> > Thanks for the answers! I will be interested to hear more about the combinatorial characterization. I agree with your intuition about regularized GBT satisfying the stronger monotonicity property. I think this is a nice paper overall.

---

> > > ### Author Response · Authors · 2025-08-04
> > > **Thank you Reviewer XoXh by Authors**
> > >
> > > We would like to thank the reviewer for the thorough and constructive feedback provided. We are glad that you found this work interesting, and deemed it acceptable for this conference. We hope to share our combinatorial characterization soon, hopefully in the coming months!
> > >
> > > Best regards, Authors

---

### Official Review · Reviewer_Ag91 · 2025-07-03

**Clarity:** 2
**Significance:** 2
**Originality:** 3
**Rating:** 4
**Confidence:** 2

**Summary:**

The paper proposes a Generalised Linear Bradley-Terry model with a specific set of embeddings they name - diffusion embeddings. The paper proves that these embeddings result in monotonic behaviour during preference learning.

**Questions:**

See Weaknesses above

**Ethical Concerns:**

["NO or VERY MINOR ethics concerns only"]

**Final Justification:**

The authors have addressed my concerns, committed to improving the clarity of the motivation in the introduction and included further empirical evidence in line with other reviewer concerns. For these reasons I increase my score from a 3 to a 4.

**Limitations:**

The authors highlight the constraint that the proposed method is dependent upon diffusion embeddings.

**Quality:**

3

**Strengths And Weaknesses:**

**Strengths**

- The setting studied by the authors is novel to the best of my knowledge.
- The paper lays out rigorous theoretical proofs of its claims and supports these with experimental results.
- The work is mostly clear and easy to follow.

**Weaknesses**

I recommend a weak reject at the moment because I do not believe the motivation for the work aligns with the method proposed.

- The authors motivate the work through DPO, GPO and other RLHF works that focus on alignment within the LLM literature, however I am uncertain if the results presented in this paper align with that motivation. The authors do not present any experiments on an LLM setting and the class of problems considered in this work is not easily extended to the class of LLM models, as to the best of my knowledge the setting is akin to tabular with an embedding for each alternative A. I may well have misunderstood this aspect of the work though - thus my low confidence score.
- The final paragraph of the introduction states that this work finds direct applications within the collaborative scoring of social media content - I am unfamiliar with this setting but would be keen to understand more as the work may present better when motivated in this context.
- The experiments lack baselines against other preference optimization work - particularly other works from the collaborative content social media content literature would enrich the work and provide further comparisons to evaluate the method against.
- Some clarity issues in the paper - in line 185 the regularisation $\mathcal{R}$ overrides the notation for the comparison value defined in line 107.

---

> ### Author Rebuttal · Authors · 2025-07-31
>
> We thank the reviewer for the time they took to review our paper, and for their remarks.
>
> > The authors motivate the work through DPO, GPO and other RLHF works that focus on alignment within the LLM literature, however I am uncertain if the results presented in this paper align with that motivation. The authors do not present any experiments on an LLM setting and the class of problems considered in this work is not easily extended to the class of LLM models, as to the best of my knowledge the setting is akin to tabular with an embedding for each alternative A.
>
> We apologize if our paper gave the impression that we would focus on LLM.
> We only mentioned DPO/RLHF/LLM to highlight (1) the growing importance of preference learning in AI, and (2) the bizarre features of many of today's preference learning deployments, such as in the case of LLM.
> The focus of our paper is given by line 46, which does not include language model alignment.
> Although the generalization may prove useful of the alignement of AI models in the long run, our motivation stems from the direct applications that GBT and its extensions find in social media content recommendation, as discussed in the original GBT paper and the related work [13, 14]. In particular, the motivation for alterantives to current social media recommender systems is provided in [14].
>
> > The final paragraph of the introduction states that this work finds direct applications within the collaborative scoring of social media content - I am unfamiliar with this setting but would be keen to understand more as the work may present better when motivated in this context.
>
> The main references we give are [13] and [14], which refer to a project called Tournesol.
> They already reportedly use the generalized Bradley-Terry model, which guarantees monotonicity but not generalization.
> We believe that our work could help this project improve their recommendation systems which are based on community preferences, without sacrificing monotonicity.
>
> > The experiments lack baselines against other preference optimization work - particularly other works from the collaborative content social media content literature would enrich the work and provide further comparisons to evaluate the method against.
>
> To the best of our knowledge, GBT and BT are the only comparison-based preference learning models that provide monotonicity guarantees. Building models from comparisons explicitly provided by users seems a highly desirable property in social media content literature. We thus do not consider the usual recommender system's litterature, which are the backbone of the current social media, and rely on data that is not explicitly provided by users.
>
> > Some clarity issues in the paper - in line 185 the regularisation $\mathcal R$ overrides the notation for the comparison value defined in line 107.
>
> We have fixed right away this notation issue. Thank you.
>
> We hope to have addressed all your concerns. We remain at your disposal may you have any further questions or require additional information. We would be grateful if you could consider revising your score based on the answers we provided.

---

> > ### Author Response · Authors · 2025-08-06
> > **Please consider engaging in the discussion**
> >
> > We thank the reviewer for their review and encourage you to engage in the discussion, replying to our rebuttal.
> >
> > We have carefully addressed the main concerns in detail, and updated the manuscript accordingly. Is there any remaining concern before you can consider increasing your score? We would be glad to clarify any further concerns, if any.
> >
> > Best regards,
> > Authors

---

> ### Comment · Reviewer_Ag91 · 2025-08-07
>
> I’d like to thank the authors for their detailed answer to my review, and patience in awaiting my response. I raise my score to recommend the paper is accepted.
>
> > The focus of our paper is given by line 46, which does not include language model alignment.
>
> Thank you for clarifying, this addresses my concern about the lack of LLM experiments and the applicability of this method to these settings. Given reviewer **yQmu** seemed to reach a similar conclusion I suspect the authors should spend further effort revising the introduction to make this distinction as clear as possible.
>
> > The experiments lack baselines against other preference optimization work
>
> I have similar concerns to reviewer ‘**DMrc’** in terms of the thorough experimental evaluation of this method, on real world data and with comparison to baselines beyond GBT. The authors have addressed these concerns suitably within their rebuttal to DMrc.
>
> For these reasons I raise my score.

---

> > ### Author Response · Authors · 2025-08-08
> > **Thank you Reviewer Ag91**
> >
> > Following your suggestion and reviewer yQmu, we will revise the introduction to improve clarity regarding its applications to LLM finetuning and social media content recommendation. We will also include the plot for GBT and GBTD performance on real-world data in the appendix, as discussed with reviewer DMrc.
> >
> > We would like to thank the reviewer for the thorough and constructive feedback provided. We are glad that you found this work interesting, and deemed it acceptable for this conference.
> >
> > Best regards,
> > Authors

---

### Official Review · Reviewer_yQmu · 2025-07-06

**Clarity:** 2
**Significance:** 2
**Originality:** 3
**Rating:** 4
**Confidence:** 4

**Summary:**

This paper addresses the overlooked issue of monotonicity in preference learning, where models should assign higher value to preferred alternatives. While existing models like LLMs often violate this, the paper introduces a novel class of Linear Generalized Bradley-Terry models with Diffusion Priors, which ensure monotonicity under specific embedding conditions and can generalize to unseen comparisons. Experiments confirm improved accuracy, especially in low-data regimes.

**Questions:**

Q1. To provide more convincing real-world numerical evidence, can the experiments be designed and operated at an industrial level, e.g., using UltraFeedback or Anthropic-HH datasets?

Q2. Generalisation of BT model has been discussed in various formats and contexts.

In many real-world datasets, binary reward models, such as the Bradley-Terry (BT) model, have been known to be vulnerable to 'intransitivity' risk because they rely on scalar variables, assuming all preferences are transitive.
- The literature below studies representative preference datasets in the real world, where the 'transitive' relationship between preference annotations may not always hold, and presents a generalized version of the BT model that considers the intransitivity of human preferences.
- https://arxiv.org/abs/2409.19325 (Duan et al, 2017) : Quantitative investigation into the importance of generalization and a generalized model that focuses on explainable representation learning.
- 10.1109/ICPR48806.2021.9412462 (Gu et al, 2021): A low-rank matrix approach for the generalization.

**Ethical Concerns:**

["NO or VERY MINOR ethics concerns only"]

**Final Justification:**

The authors have provided a candid response and meaningful motivation of the paper. The perspective of 'generalization of preference learning models with monotonicity' is essential as a scientific area for decision science and the modern LLM industry. I will look forward to seeing a deep dive into this topic in the future; therefore, I raise my score to Accept.

**Limitations:**

yes

**Paper Formatting Concerns:**

No issue is identified regarding the submitted paper content, paper references, the NeurIPS paper checklist.

**Quality:**

3

**Strengths And Weaknesses:**

Strength:
1. This paper focuses on a neglected but important data-driven issue in LLM fine-tuning, that is, the generalization of reward evaluations.
2. Theoretical contributions are valid and identify conditions under which monotonicity can be guaranteed in linear comparison-based models with diffusion priors.

Weakness:
1. The experiments are not organized at an industrial level. The numerical evidence is on a miniature scale, insufficient for technical audiences.

---

> ### Author Rebuttal · Authors · 2025-07-31
>
> We thank the reviewer for the time they took to review our paper, and for their remarks.
>
> > Q1. To provide more convincing real-world numerical evidence, can the experiments be designed and operated at an industrial level, e.g., using UltraFeedback or Anthropic-HH datasets?
>
> Our models apply to any problem that can be formulated in terms of pairwise preferences between alternatives, alternatives for which we want to infer scores (monotonically). Since both datasets provide comparisons, our models could, in principle, be applied to infer scores for the chosen responses given a prompt.
>
> However, these datasets have been designed for LLM applications, where the goal is not so much about inferring scores for the chosen responses, but more about computing the probability of generating a response given a prompt. It is expected that human feedback can be leveraged to adjust these probabilities in the "expected way". Monotonicity is one formalization of this "expectation".
>
> We believe that monotonicity of LLMs should be studied more thoroughly. As mentioned in our introduction, the fact that the models referred in [25,30,1] fail to guarantee monotonicity is one of the core motivations behind the present work. We chose to focus, in this paper (line 46), on models that are well-structured, and simple enough to analyze, for which we *can* provide monotonicity guarantees. Given the complexity of LLMs, we reserve the study of their monotonicity for distinct work.
>
> > Q2. Generalisation of BT model has been discussed in various formats and contexts. In many real-world datasets, binary reward models, such as the Bradley-Terry (BT) model, have been known to be vulnerable to 'intransitivity' risk because they rely on scalar variables, assuming all preferences are transitive. The literature below studies representative preference datasets in the real world, where the 'transitive' relationship between preference annotations may not always hold, and presents a generalized version of the BT model that considers the intransitivity of human preferences. https://arxiv.org/abs/2409.19325 (Duan et al, 2017) : Quantitative investigation into the importance of generalization and a generalized model that focuses on explainable representation learning. 10.1109/ICPR48806.2021.9412462 (Gu et al, 2021): A low-rank matrix approach for the generalization.
>
> We focus our attention on monotonicity of scores for various BT-related models. Indeed, since these probabilistic models infer *scores*, they implicitly assume that intransitive judgements are due to noise. We do have an ongoing line of research that consider approaches that try to better model human decision-making. We were not aware of the references you mention, will include them in the related work, and consider them for our own research. Thank you.
>
> We hope to have addressed all your concerns. We remain at your disposal may you have any further questions or require additional information. We would be grateful if you could consider revising your score based on the answers we provided.

---

> > ### Comment · Reviewer_yQmu · 2025-08-03
> >
> > I thank the authors for their candid response and elaboration of the motivation. The perspective of 'generalization of preference learning models with monotonicity' is essential as a scientific area for decision science and the modern LLM industry. I will look forward to seeing a deep dive into this topic in the future; therefore, I raise my score to Accept.

---

> > > ### Author Response · Authors · 2025-08-04
> > > **Thank you Reviewer yQmu by Authors**
> > >
> > > We would like to thank the reviewer for the thorough and constructive feedback provided. We are glad that you found this work interesting, and deemed it acceptable for this conference. We also look forward further developments in preference learning!
> > >
> > > Best regards,
> > > Authors

---

### Comment · Area_Chair_Fgx8 · 2025-08-01
**Author-Reviewer Discussions (July 31 - Aug 6)**

Dear reviewers,

Authors have provided their rebuttals and now we are in Author-Reviewer Discussions (July 31 - Aug 6) period. Please read authors' rebuttal and give comments. Thanks!

---

### Note · Authors · 2025-08-15

We thank the Area Chair for the time and help in having our work reviewed for the conference.

We appreciate that the reviewers found our work makes a relevant contribution to the field of preference learning. We summarize below the **strenghts**, **concerns**, and how the **discussion** led to improvements.

Recognized **strengths**
- Novel, rigorous theoretical contribution, supported with experimental results
- The work is mostly clear and easy to follow
- 'Generalization of preference learning models with monotonicity' is essential as a scientific area for decision science and the modern LLM industry
- Extending GBT to allow score-generalization is potentially impactful for the community

Reviewers' **concerns**
- Some reviewers were confused about the mention of RLHF/DPO in the introduction, since the paper does not study preference learning for LLMs specifically.
-> We responded that our point there was to emphasize that monotonicity is both non-trivial and relevant in important applications such as LLM finetuning or social media content recommendation.
  We argued that the present work focuses on a model that is both monotone and able to generalize. We updated our paper to clarify our intentions.
- Experiments are not organized at an industrial level, e.g. using UltraFeedback or Anthropic-HH datasets
-> We answered that these datasets are designed for LLM applications, which involve their own specific details. Monotonicity of LLMs is an important research direction, which we consider, though, out of the scope of the paper. We will add to the appendix a comparison of GBTD and GBT on real-world, social-media recommendation data.
- Experiments do not compare against other preference optimization frameworks
-> We clarified, BT and GBT are the only models that have monotonicity guarantees (RLHF & DPO do not), and rely on data explicitly provided by users (classical recommender systems do not).

Improvements after **discussion**
- Revised introduction to clarify applications to LLM finetuning and social media content recommendation.
- Clarified our notations.
- Added error bars in plots.
- Include in the appendix a plot that compares performance of GBTD vs GBT on real-word data (with a precise description of the experiment, and update to the github repository).

We did our best to respond in detail to all the reviewers comments; we hope this summary will help you in the assessment of the discussion. Thanking you again

---

### Decision · Program_Chairs · 2025-09-17

**Decision:**

Accept (poster)

**Comment:**

In this paper, authors advance the understanding of the set of models with generalization ability that are monotone, i.e., authors proposed a new class of Linear Generalized Bradley-Terry models with Diffusion Priors, and identify sufficient conditions on alternatives' embeddings that guarantee monotonicity. Experimental results verified the effectiveness of the proposed method.

The final rating of this paper is 2 borderline accept, 2 accept.

The strength of this paper given by reviewers are:
1) paper solved an important data-driven issue in LLM fine-tuning (Reviewer yQmu, XoXh)
2) solid contribution (Reviewer yQmu, Ag91, XoXh)
3) the setting is novel and interesting. (Reviewer Ag91, DMrc)
4) the work is clear and easy to follow (Reviewer Ag91, XoXh)

The weaknesses are:
1) experiments are not organized at an industrial level (Reviewer yQmu)
2) do not believe the motivation for the work aligns with the method proposed. (Reviewer Ag91)
3) A few minor points that could be made clearer (Reviewer XoXh)
4) writing could be improved (Reviewer DMrc)
5) experimental evaluation (Reviewer DMrc)

After rebuttal, reviewer yQmu, Ag91 convinced by authors and gave a borderline accept rating. Reviewer XoXh, DMrc  still had some minor concerns but think this is a nice paper and give accept rating.

Given all these AC decided to accept this paper.